# Deciphering the molecular mechanisms of FET fusion oncoprotein–DNA hollow co-condensates

Linyu Zuo[1,7], Qirui Guo[1,7], Cheng Li[1,7], Kecheng Zhang[2], Yancao Chen[1], Baiyi Jiang[1], Zhixing Chen ®[2,3], Yufei Xia ®[4] ✉, Long Qian ®[1] ✉, Lei Zhang ®[1,5] ✉ & Zhi Qi ®[1,6] ✉

Biomolecules such as nucleic acids and proteins can undergo phase separation to form biomolecular condensates with diverse architectures. Here, we report that the FUS/EWS/TAF15 family fusion oncoprotein FUS-ERG forms hollow co-condensates with double-stranded DNA containing GGAA microsatellites. Through a combination of biochemical assays, super-resolution imaging, and mathematical modeling, we reveal that the interior surface of hollow co-condensates exhibits properties distinct from those of the external surface, a phenomenon we term nested asymmetric phase separation. Furthermore, we harness FUS-ERG for DNA-based information manipulation and demonstrated the hollow condensate morphology uniquely enhances data sorting specificity, enabling targeted DNA deletion within dsDNA libraries and facilitating dynamic, hierarchical data selection. These findings provide critical insights into the biophysical mechanisms underlying multicomponent phase-separated cellular bodies and establish a foundation for leveraging condensate morphology in biotechnology.

Phase separation is a fundamental physicochemical process by which biomolecules—such as proteins and nucleic acids—spontaneously demix from the surrounding solution (the dilute phase) to form concentrated, mesoscale condensates (the dense phase) within cells[1–5]. These biomolecular condensates are dynamic, membraneless compartments composed of multiple components and are formed independently of lipid bilayers. Their assembly is primarily driven by two mechanisms: interactions mediated by intrinsically disordered regions (IDRs) of proteins[6], and multivalent interactions among modular macromolecules[7]. Phase separation-driven condensates exhibit spatiotemporal self-organization and coarsening dynamics[8] and are integral to numerous biological processes[1,9,10]. Furthermore,

dysregulation of condensate formation has been linked to the pathogenesis of various human diseases, including neurodegenerative disorders and cancer[3].

Multicomponent condensates exhibit a range of complex architectures, including "pearl chain"-like structures[11–14], nested "Russian doll" structures[5,15], and hollow condensate architectures[16–18]. Among these, hollow condensates represent the simplest model that provides an entry point for studies of complex architectures. For example, Banerjee and colleagues[17] demonstrated the formation of hollow condensates consisting of an arginine-rich disordered nucleoprotein, protamine (PRM), in combination with RNA. They proposed that PRM and RNA assemble in a manner similar to a lipid-

[1]Center for Quantitative Biology, Academy for Advanced Interdisciplinary Studies, Peking University, Beijing, China. [2]Peking-Tsinghua Center for Life Sciences, Academy for Advanced Interdisciplinary Studies, Peking University, Beijing, China. [3]College of Future Technology, Institute of Molecular Medicine, National Biomedical Imaging Center, Beijing Key Laboratory of Cardiometabolic Molecular Medicine, Peking University, Beijing, China. [4]Key Laboratory of Biopharmaceutical Preparation and Delivery, Chinese Academy of Sciences, Beijing, China. [5]Beijing International Center for Mathematical Research, Center for Machine Learning Research, Peking University, Beijing, China. [6]School of Physics, Peking University, Beijing, China. [7]These authors contributed equally: Linyu Zuo, Qirui Guo, Cheng Li. ✉e-mail: yfxia@ipe.ac.cn; long.qian@pku.edu.cn; zhangl@math.pku.edu.cn; zhiqi7@pku.edu.cn

like diblock copolymer, where these copolymers, along with high concentrations of RNA or protein, lead to vesicle-like hollow condensates.

In this work, we report a new type of hollow condensates. One important FUS/EWS/TAF15 (FET) family fusion oncoprotein, FUS-ERG, can form hollow co-condensates with 25-base pair (bp) double-stranded DNA (dsDNA) containing a 4 × GGAA microsatellite sequence (25-bp 4 × GGAA dsDNA). FUS-ERG is formed by the fusion of the low-complexity domain (LCD) and the RGG domain of FUS with the DNA-binding domain (DBD) of the E26 transformation-specific (ETS) family transcription factor ERG. The DBD endowed FUS-ERG with in vivo binding specificity to a genomic microsatellite sequence characterized by GGAA repeats[19,20].

To interrogate the biophysical mechanism of FET family fusion oncoprotein hollow co-condensates and their practical implications, we employed a multidisciplinary approach. Remarkably, super-resolution imaging experiments combined with mathematical modeling illuminated a molecular mechanism driving the formation of these hollow co-condensates, which is distinctly different from that of vesicle-like hollow co-condensates described in a previous study[17].

Biomolecular condensate offers significant advantages in facilitating rapid biomolecular self-assembly, which has been leveraged in the design of various synthetic functional structures[21,22]. Because the newly discovered FET family fusion oncoprotein hollow co-condensates involve protein-DNA interactions, we asked whether they can provide a compelling opportunity for dynamic information manipulation in DNA-based data storage. While studies have shown that in-storage DNA encapsulation within abiotic polymers enhances data longevity and PCR uniformity, DNA encapsulation has so far been produced by pre-loading in a non-selective manner and lacked dynamic storage capacity[23,24]. In this study, we leverage protein-DNA self-assembly based on sequence-specific FET fusion oncoprotein-DNA interactions and the hollow co-condensate architecture as a dynamic and selective encapsulation medium. We demonstrated in-storage precise information selection and deletion, as well as hierarchical information sorting by programming DBD-dsDNA interactions. These results underscore the potential of using multi-layer biomolecular condensate for the dynamic spatial regulation of molecular information, providing a distinctive route for in-storage molecular storage and computation.

## Results

### FUS-ERG can form hollow co-condensates with dsDNA containing GGAA microsatellite sequence

We first conducted in vitro droplet assays to determine whether FET fusion oncoprotein FUS-ERG can form biomolecular condensates. We purified a GFP-tag-labeled FUS-ERG (GFP-FUS-ERG, Supplementary Fig. 1a(i)–(ii)), and performed electrophoretic mobility shift assays (EMSAs) to validate the protein activity in vitro (Supplementary Fig. 1a(iii)). Next, biochemical assays revealed that GFP-FUS-ERG protein can form homogeneous condensate at a concentration as low as 1 μM in vitro (Fig. 1a), consistent with previous findings[25]. Mixing 0.6 μM of 25 bp dsDNA containing a random sequence (referred to as 25 bp random dsDNA) and 5 μM GFP-FUS-ERG resulted in the co-localization of both components within homogeneous droplets (Fig. 1b(i)–(ii)). However, when dsDNA substrates of the same length were designed to contain GGAA microsatellites, such as 2 × or 4 × GGAA, FUS-ERG formed hollow co-condensates with these dsDNAs (Fig. 1b(iii)–(iv)). The three-dimensional hollow architecture was confirmed by confocal microscopy (Supplementary Movie 1). Fluorescence analysis revealed that 25 bp 4 × GGAA dsDNA and GFP-FUS-ERG co-localized on the shell of hollow co-condensates (Fig. 1b(v)). Additionally, we observed two interesting phenomena. First, the homogeneous condensates formed by 25-bp random dsDNA and GFP-FUS-ERG had the dsDNA enveloped by FUS-ERG (Fig. 1b(ii)), suggesting that

the protein itself has stronger interactions with the solvent. Second, when the length of random dsDNA increased from 25 to 306 bp, hollow co-condensates can also be formed, albeit with a larger diameter (Supplementary Fig. 2a, b), suggesting that complex protein-DNA interactions govern the condensate architecture.

We investigated the conditions required for hollow co-condensate formation through in vitro droplet assays, combining 25 bp 4 × GGAA dsDNA at concentrations of 0, 0.15, 0.3, 0.6, 1.2, and 2.4 μM with GFP-FUS-ERG protein at concentrations of 0.25, 0.5, 2, 5, and 10 μM (Fig. 1c(i)). When the DNA concentration was fixed at 0.15 μM, hollow condensates formed only at a protein concentration of approximately 2 μM, with no hollow condensates observed at either lower or higher protein concentrations. At protein concentrations below 2 μM, no condensates were detected, whereas at concentrations above 2 μM, the hollow structures became filled, yielding homogeneous condensates. This non-monotonic dependence on protein concentration about the hollow structure formation is a hallmark of reentrant phase behavior[26,27].

The resulting phase diagram (Fig. 1c(ii)) reveals that the DNA-to-protein molar ratio beyond a threshold number ([DNA] / [protein] ~ 0.075) drives the hollow co-condensate formation. This threshold arises because the protein alone can form homogeneous condensates at concentrations above ~2 μM. Notably, increasing the DNA concentration from 0.15 to 2.4 μM did not alter the threshold of protein concentration for hollow structure formation, which remained fixed at ~2 μM. The molecular basis of this reentrant phase behavior in these hollow condensates remains to be elucidated and will be the subject of future investigation.

Both hollow and homogeneous condensates exhibited slow fusion kinetics (Supplementary Fig. 3a). Following this, we conducted fluorescence recovery after photobleaching (FRAP) experiments (Supplementary Fig. 3b(i)–(ii) and Methods) using GFP-FUS-ERG alone or mixed with 25 bp 4 × GGAA dsDNA. The molecular dynamics of GFP-FUS-ERG in hollow co-condensates were slower compared to homogeneous condensates, suggesting slower internal dynamics inside the shell region of hollow co-condensates.

While modifying the dsDNA sequence and length has been shown to regulate the formation of hollow condensates, we next asked how specific FUS-ERG domains affect the condensate morphology. The FUS LCD domain contains 27 [G/S]Y[G/S] repeats, where the tyrosines were shown to be important for FUS's threshold concentration for condensation[28,29]. When nine relevant tyrosine residues were changed to serine in GFP-FUS-ERG (9YS) (Supplementary Fig. 1c), the threshold concentration for condensation remained unchanged (Fig. 1d(i)). However, 5 μM GFP-FUS-ERG (9YS) cannot form hollow co-condensates with 25 bp 4 × GGAA dsDNA (Fig. 1d(ii)). When all 27 tyrosines were mutated to serine in GFP-FUS-ERG (27YS) (Supplementary Fig. 1d), this mutant cannot form condensates even at 20 μM (Fig. 1d(iii)), and 5 μM GFP-FUS-ERG (27YS) with dsDNA also failed to form condensates (Fig. 1d(iv)).

For the FUS RGG motif, we observed a similar trend. Mutation of five key arginine residues to alanine (GFP-FUS-ERG (5RA), Supplementary Fig. 1e) did not affect the threshold concentration for condensation (Fig. 1e(i)). In contrast, mutating nine arginine residues to alanine (GFP-FUS-ERG (9RA), Supplementary Fig. 1f) significantly increased the threshold concentration for condensation (Fig. 1e(iii)). GFP-FUS-ERG (5RA) formed both homogeneous and hollow co-condensates with 25 bp 4 × GGAA dsDNA (Fig. 1e(ii)). Conversely, GFP-FUS-ERG (9RA) only formed smaller homogeneous condensates with 25-bp 4 × GGAA dsDNA (Fig. 1e(iv)). Lastly, GFP-FUS-DDIT3, another FET fusion oncoprotein, has been reported to form similar spherical shell in vivo[30]. We repeated this experiment in U2OS cells (Supplementary Fig. 4a). Interestingly, when all arginine residues in the RGG motif were mutated to alanine, FUS-DDIT3-GFP (9RA) also cannot form this architecture in vivo (Supplementary Fig. 4b). Taken together,

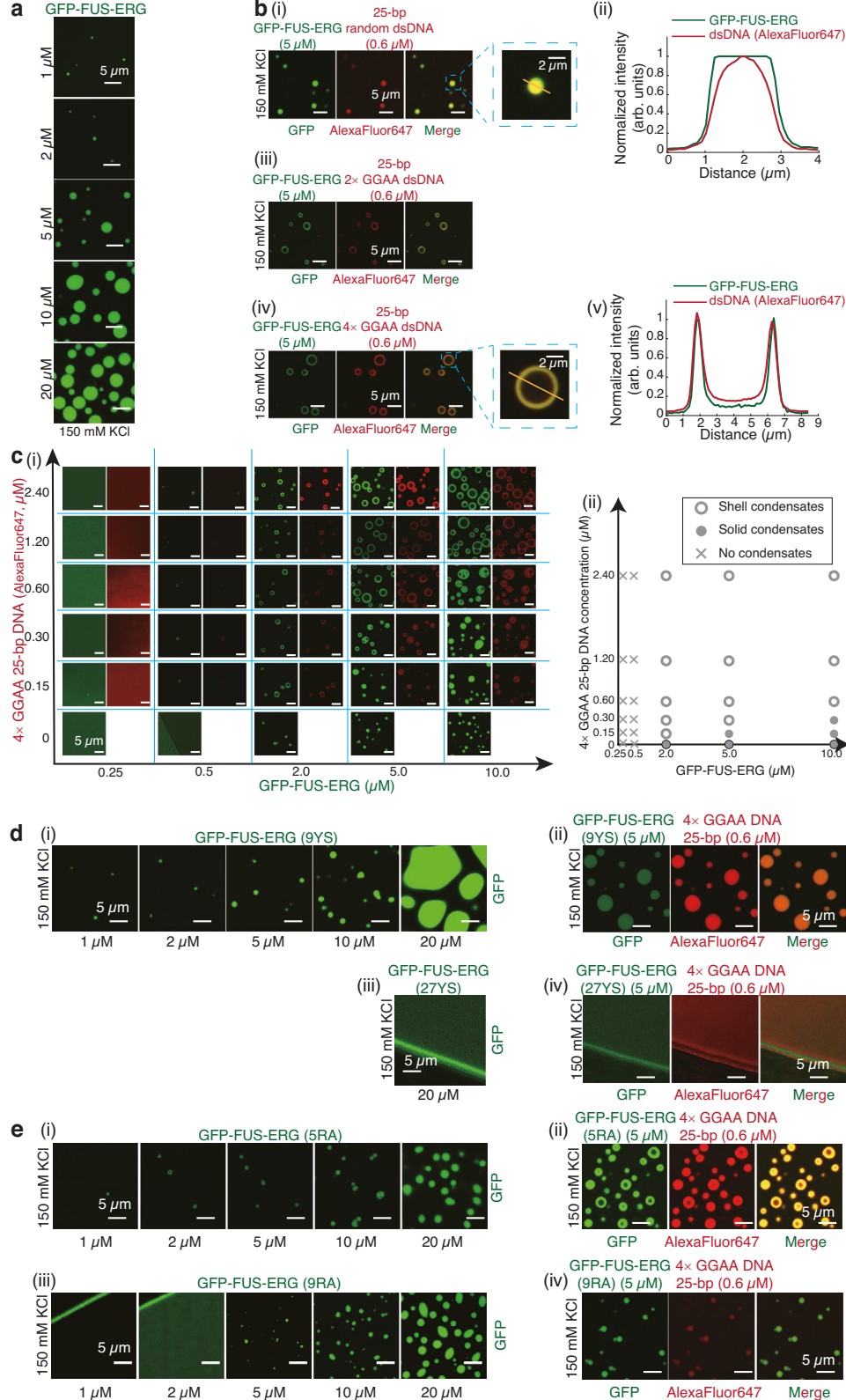

these results indicate that the LCD and RGG motifs in FUS-ERG strongly regulate hollow co-condensate formation.

Next, we sought to track the temporal dynamic of condensate formation by a two-step experiment. Initially, we used GFP-FUS-ERG to form homogeneous condensates (Fig. 1a). Subsequently, we introduced 25 bp 4 × GGAA dsDNA (Fig. 2a(i), time 0). Remarkably, the dsDNA substrates promptly enveloped the exterior surface of the homogeneous GFP-FUS-ERG condensates (Fig. 2a(v)). This process was monitored continuously for 90 min (Fig. 2a and Supplementary Movie 2). By 30 min (Fig. 2a(ii)), the dsDNA substrates had completely infiltrated the condensate, coinciding with a reduction in protein intensity within the central region (Fig. 2a(vi)). By 90 min (Fig. 2a(iv)), hollow co-condensates had fully formed, with both dsDNA and protein concentrated on the shell of the hollow co-condensates (Fig. 2a(vii)).

**Fig. 1 | FUS-ERG can form hollow co-condensates with dsDNA containing GGAA microsatellite sequence. a** GFP-FUS-ERG can form biomolecular condensate in the concentration of 1, 2, 5, 10 and 20 μM. **b** 5 μM GFP-FUS-ERG mixed with 0.6 μM 25-bp random dsDNA (i), 0.6 μM 25 bp 2 × GGAA dsDNA (iii); 0.6 μM 25 bp 4 × GGAA dsDNA (iv). dsDNA was labeled with AlexaFluor647. (ii) and (v) Normalized intensity profiles in (i) and (iv). **c** (i) GFP-FUS-ERG mixed with 25 bp 4 × GGAA dsDNA; (ii) Phase diagram in (i). **d** (i) GFP-FUS-ERG (9YS) can form biomolecular condensate in the concentration of 1, 2, 5, 10, and 20 μM; (ii) 5 μM GFP-FUS-ERG (9YS) mixed with 0.6 μM 25-bp 4 × GGAA dsDNA. (iii) GFP-FUS-ERG (27YS) cannot form biomolecular condensate in 20 μM; (iv) 5 μM GFP-FUS-ERG (27YS) mixed with 0.6 μM 25-bp 4 × GGAA dsDNA. **e** (i) GFP-FUS-ERG (5RA) can form biomolecular condensate in the concentration of 1, 2, 5, 10, and 20 μM; (ii) 5 μM GFP-FUS-ERG (5RA) mixed with 0.6 μM 25 bp 4 × GGAA dsDNA. (iii) GFP-FUS-ERG (9RA) can form biomolecular condensate in the concentration of 5, 10, and 20 μM; (iv) 5 μM GFP-FUS-ERG (9RA) mixed with 0.6 μM 25-bp 4 × GGAA dsDNA. All in vitro droplet assays were executed under physiological conditions, specifically 40 mM Tris-HCl (pH 7.5), 150 mM KCl, 2 mM MgCl₂, 1 mM DTT and 0.2 mg/mL BSA, with thorough mixing and a 30-minute incubation period prior to imaging, unless otherwise indicated. Scale bar: 5 μm in (**a**, **b**(i), (iii), (iv), **c**(i), **d**, and **e**). Scale bar: 2 μm in **b**(i) insert and **b**(iv) insert. Source data are provided as a Source Data file.

We conducted two control experiments. First, when 25 bp random dsDNA was used, no hollow co-condensates formed even after 90 min (Fig. 2b), and significantly fewer dsDNA molecules were transferred into the GFP-FUS-ERG condensates (Fig. 2c). Second, we repeated the experiment shown in Fig. 2a, substituting the 25 bp 4 × GGAA dsDNA with 306 bp random dsDNA (Supplementary Fig. 2c). Interestingly, even after 90 min (Supplementary Fig. 2c(iv)), the dsDNA infiltrated the condensates and colocalized with protein, but no hollow condensate formation was observed—contrasting sharply with the outcome in Fig. 2a. These findings demonstrate the spontaneous transfer of dsDNA containing GGAA microsatellites into the condensates accompanying hollow co-condensate formation. These findings also highlight a key mechanistic distinction between site-specific binding by microsatellite DNA and non-specific adsorption or wetting by long random DNA[31], which may differentially stabilize hollow condensates. Exploring this distinction represents an interesting avenue for future research.

## The formation of hollow co-condensates is driven by nested asymmetric phase separation

To uncover the molecular mechanism driving the formation of hollow co-condensates, we focused on the processes that define the development of the external and internal surfaces. GFP-FUS-ERG alone forms biomolecular condensate with the working buffer (Fig. 1a), suggesting that the stable external surface arises from interactions between free proteins and the buffer. Furthermore, in co-condensates formed by GFP-FUS-ERG and 25 bp random dsDNA, the GFP-FUS-ERG boundary extended beyond the dsDNA boundary (Fig. 1b(i)–(ii)). This observation strongly indicates that phase separation between free proteins and the surrounding buffer could drive the formation of a stable external surface.

To elucidate the formation of the internal surface, we firstly employed RNA probes to discern the internal characteristics of the hollow co-condensates. Specifically, we combined SNAP-tag-labeled FUS-ERG (SNAP-FUS-ERG, Supplementary Fig. 1b(i)–(iii)) with 25 bp 4 × GGAA dsDNA to induce the formation of hollow co-condensates (Supplementary Fig. 1b(iv)). After a 30 min incubation with Poly-U RNA, we observed the penetration of the RNA through the shell, resulting in their enrichment within the lumens of the hollow co-condensates (Fig. 3a, d). In contrast, RNA encountered difficulty in penetrating the homogeneous condensates formed by SNAP-FUS-ERG alone (Fig. 3c, d) or in conjunction with 25-bp random dsDNA (Fig. 3b, d). As the RGG motif (amino acids 485–539) (Supplementary Fig. 1a(i)) is the only domain within GFP-FUS-ERG capable of tightly binding RNA, these findings strongly suggest that the internal surface of the hollow condensates was aligned with RGG motifs whereas in homogeneous condensates the RGG motifs were likely sequestered within the protein.

Integrating the RNA probe experiments (Fig. 3a–d) with earlier observations of the translocation of GGAA motif-containing dsDNA into the condensates (Fig. 2a), we propose an interesting hypothesis for the formation of the internal surface. When a dsDNA substrate containing GGAA motifs binds to a free FUS-ERG molecule near the condensate's exterior boundary (Fig. 2a(i)), a protein–dsDNA complex is formed, inducing a conformational change in the protein from a Closed state, where the RGG motif is buried and only display week affinity to RNA, to an Open state where the RGG motif is exposed (Fig. 3e(i)). The results shown in Fig. 2 suggest that this protein–dsDNA complex can translocate into the condensate's inner region, where the exposed hydrophilic RGG motif (Supplementary Fig. 5a) accumulates, thereby establishing the internal surface (Fig. 3e(ii)).

To test the hypothesis of conformational switch from closed to opened state of GFP-FUS-ERG in Fig. 3e, we divided the full-length GFP-FUS-ERG into two separate modules: GFP-FUS-LCD and RGG-ERG (comprising the RGG motif and ERG domain). SDS-PAGE and EMSA analyses confirmed that both proteins were purified to high homogeneity and that RGG-ERG retained sequence-specific DNA-binding activity (Supplementary Fig. 6a). Notably, GFP-FUS-LCD did not exhibit any detectable DNA-binding, even at very high concentrations (Supplementary Fig. 6b).

To test whether FUS-ERG can adopt a closed conformation mediated by interactions between the FUS-LCD and the ERG domain, we performed an in vitro droplet assay. GFP-FUS-LCD and RGG-ERG proteins were incubated either individually or in combination at equimolar concentrations for 30 minutes at room temperature prior to imaging (Supplementary Fig. 7a(i)). GFP-FUS-LCD alone did not form condensates, while RGG-ERG readily formed droplets on its own. Upon mixing, the two proteins exhibited clear co-localization of fluorescent signals (Supplementary Fig. 7b(i)), indicating direct interactions between GFP-FUS-LCD and RGG-ERG. These results strongly support the presence of a closed state driven by intramolecular or intermolecular contacts between the FUS-LCD and ERG domain (Fig. 3e(i)).

To determine whether dsDNA containing GGAA motifs can induce a conformational transition of FUS-ERG from a closed to an open state, we performed an in vitro droplet assay in which GFP-FUS-LCD and RGG-ERG were mixed at equimolar concentrations and incubated with increasing amounts of 25 bp 4 × GGAA DNA or control 25 bp random DNA (Supplementary Fig. 7a(ii–iii)). At a low concentration of 4 × GGAA DNA (0.3 μM), GFP-FUS-LCD continued to co-localize with RGG-ERG. However, at or above 0.6 μM, this co-localization was abolished (Supplementary Fig. 7b(ii)). In contrast, random DNA failed to disrupt co-localization even at concentrations as high as 2.4 μM (Supplementary Fig. 7b(iii)). These results indicate that only high concentrations of GGAA-containing dsDNA can displace GFP-FUS-LCD from RGG-ERG condensates, likely by disrupting the interaction between the FUS-LCD and ERG domain. This observation aligns with the previous work[32], which showed that high-affinity DNA disrupts the interaction between EWS-LCD and FLI1-DBD within condensates. Collectively, these data strongly support that GGAA motif-containing dsDNA induces a transition of FUS-ERG from a closed to an open state. Moreover, our results demonstrate that control dsDNA lacking GGAA motifs does not elicit this conformational change. Together with the droplet assay results from Supplementary Fig. 7 and Fig. 1b, our findings support that the closed-to-open transition of FUS-ERG is a critical prerequisite for the formation of hollow condensates.

Despite the above experiments, we are still unable to precisely demonstrate the existence of conformational changes at the molecular level. If this hypothesis holds true, the interior surface of the hollow co-condensates is expected to exhibit the highest concentration of

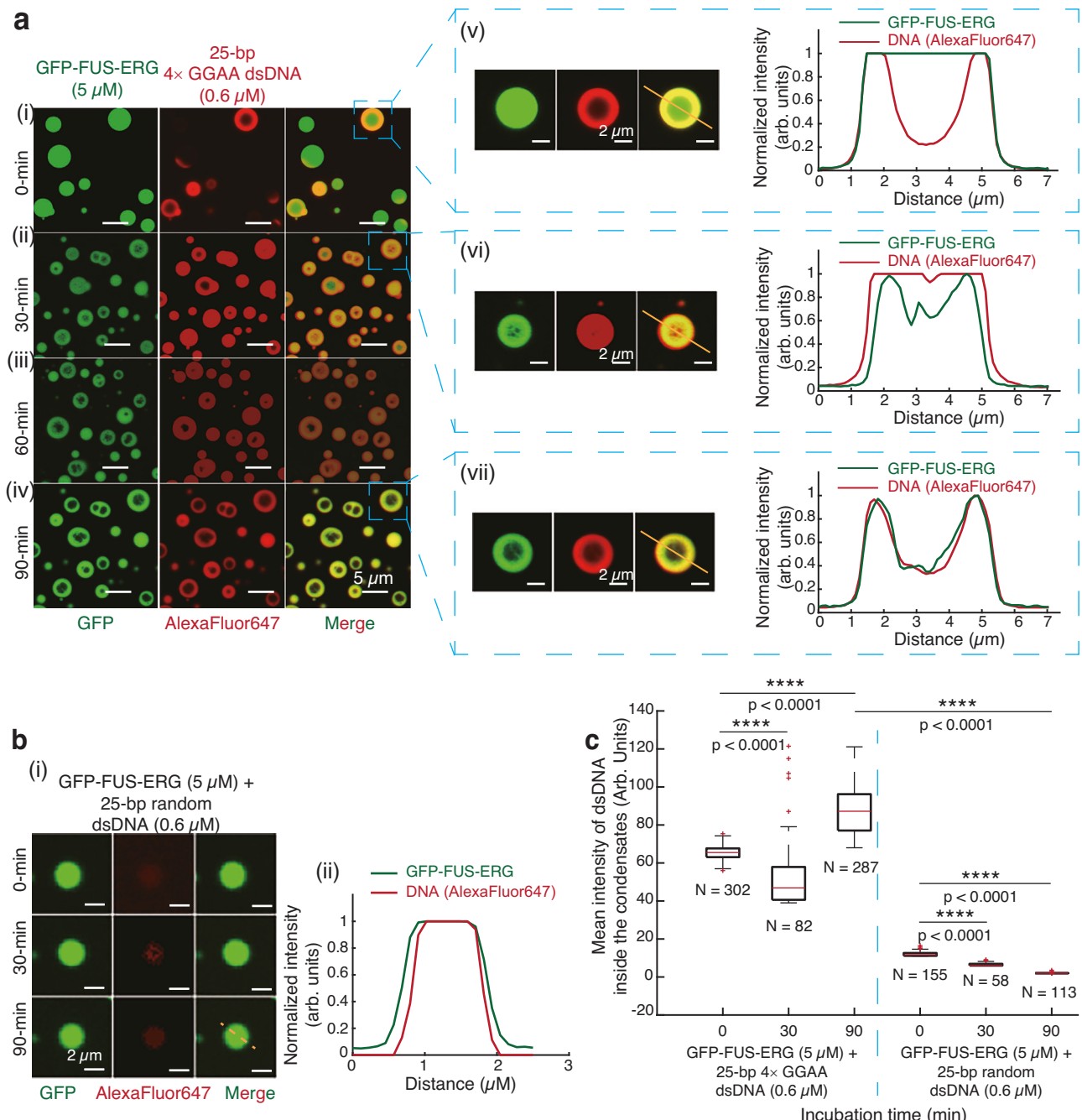

**Fig. 2 | 25-bp dsDNA substrates containing GGAA microsatellites can transfer into the homogeneous FUS-ERG condensates, inducing the hollow co-condensate formation. a** Time course of hollow co-condensate formation of 5 µM GFP-FUS-ERG mixed with 0.6 µM AlexaFluor647-labeled 25 bp 4 × GGAA dsDNA at 0 min (i), 30 min (ii), 60 min (iii), and 90 min (iv). At the 0 min time point, we injected dsDNA. (v), (vi), and (vii) are representative events of hollow co-condensate formation and normalized intensity profiles from (i), (ii), and (iv). **b** (i) Time course of hollow co-condensate formation of 5 µM GFP-FUS-ERG mixed with 0.6 µM AlexaFluor647-labeled 25-bp random dsDNA at 0, 30, and 90 min. At the 0 min time point, we injected dsDNA. (ii) is the normalized intensity profile at the 90 min time point. **c** Boxplot of the mean intensity of dsDNA inside the condensates for GFP-FUS-ERG with 25 bp random dsDNA and 25 bp 4 × GGAA dsDNA. The total

number N examined over one-time in vitro droplet experiments. For the boxplot, the red bar represents median. The bottom edge of the box represents 25th percentiles, and the top is 75th percentiles. Most extreme data points are covered by the whiskers except outliers. The '+' symbol is used to represent the outliers. Statistical significance was analyzed using unpaired t test for two groups. P value: two-tailed; p value style: GP: 0.1234 (ns), 0.0332 (*), 0.0021 (**), 0.0002 (***), < 0.0001 (****). Exact P values are as follows: $P < 0.0001$ for all conditions: 4 × GGAA dsDNA 0 vs. 30 min; 4 × GGAA dsDNA 0-min vs. 90 min; 4 × GGAA dsDNA 90 min vs. random dsDNA 90 min; random dsDNA 0 min vs. 30 min; random dsDNA 0 vs. 90 min. Confidence level: 95%. Scale bar: 5 µm in a(i)–(iv). Scale bar: 2 µm in a(v)–(vii) and b(i). Source data are provided as a Source Data file.

dsDNA–protein complexes, with a radial decrease in concentration toward the exterior. To investigate this distribution, we employed Stimulated Emission Depletion (STED) microscopy, which provides sub-100 nm resolution[33], to image the dsDNA distribution within

hollow condensates. Fluorescent signals from the ATTO647N-labeled (1:100) dsDNA substrates are presented in Fig. 3f(i). Quantitative imaging analysis (Supplementary Methods) revealed that the interior surface exhibited significantly higher dsDNA intensity, with a gradual

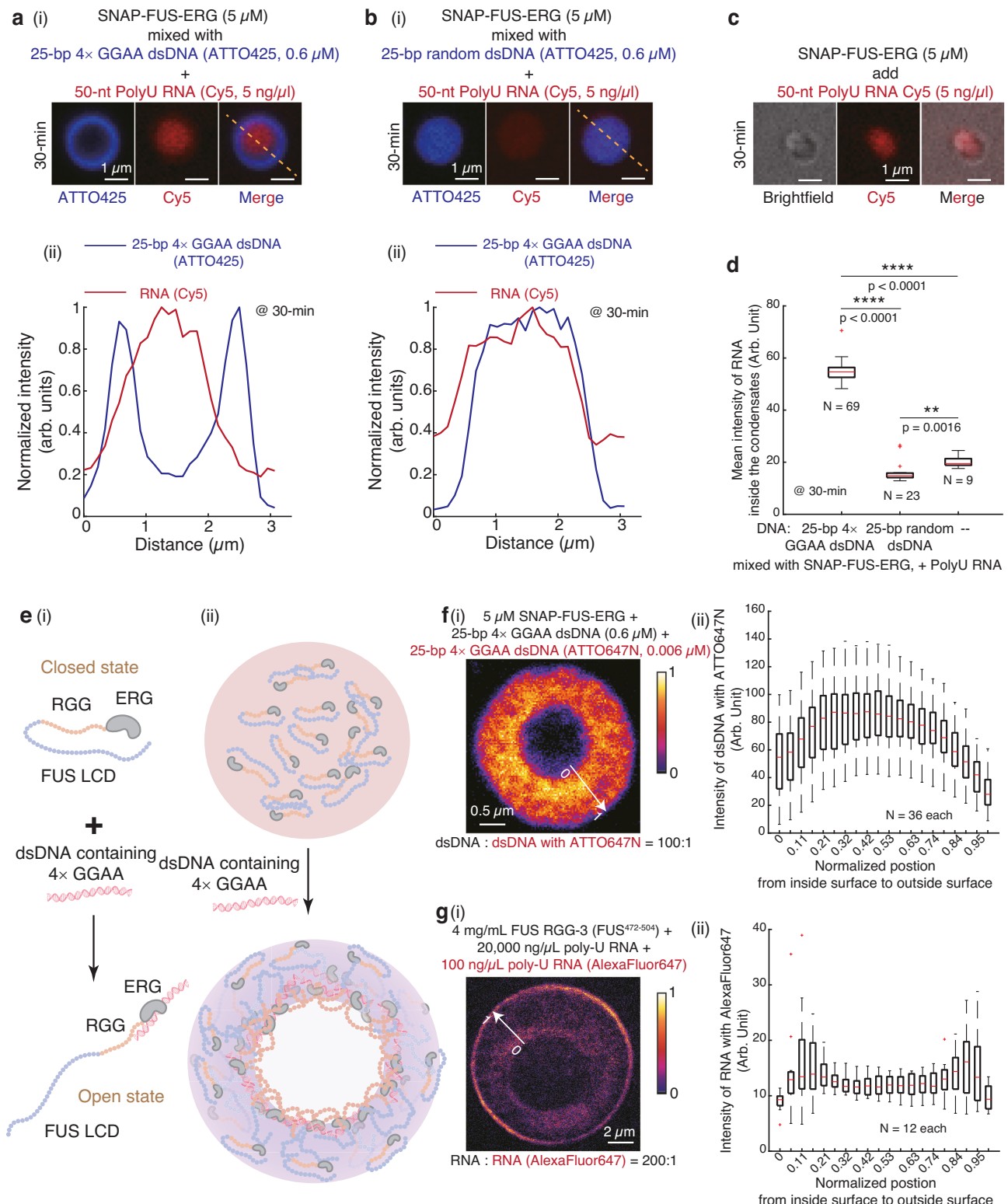

decrease in intensity along the radial axis toward the exterior (Fig. 3f(ii)), confirming the asymmetric distribution of the FUS-ERG–dsDNA complex.

The hollow co-condensates formed by PRM and RNA[17] are expected to exhibit a symmetric distribution of RNA due to their vesicle-like characteristics. As the RGG motifs of FUS are similar to PRM in RNA binding abilities, we repeated the STED experiment to image the distribution of Poly-U RNA co-condensates with the third RGG

motif inside FUS (FUS RGG-3, 471-504) into vesicle-like droplets (Supplementary Fig. 8). The fluorescent signals of RNA substrates are depicted in Fig. 3g(i), and subsequent image analysis confirmed the symmetric distribution of RNA substrates (Fig. 3g(ii)). Collectively, these super-resolution imaging results elucidate a molecular mechanism underlying the internal surface formation of FUS-ERG and dsDNA hollow co-condensates, which we term "nested asymmetric phase separation".

**Fig. 3 | RNA probe and STED imaging reveal the molecular mechanism of hollow co-condensate formation. a−c** 5 μM SNAP-FUS-ERG first mixed with 0.6 μM ATTO425-labeled 25-bp 4 × GGAA dsDNA (a(i)), random dsDNA (b(i)), or no dsDNA (c). Next, 5 ng/μL 50-nt Cy5-labeled PolyU RNA was injected. a(ii) and b(ii) are normalized intensity profiles from a(i) and b(i). **d** Boxplot of the mean intensity of RNA inside the condensates in a(i), b(i) and c. The total number N examined over three-time in vitro droplet experiments. Exact P values are as follows: 4 × GGAA dsDNA vs. random dsDNA, $P < 0.0001$; 4 × GGAA dsDNA vs. no dsDNA, $P < 0.0001$; random dsDNA vs. no dsDNA, $P = 0.0016$. **e** Schematics of hollow co-condensate formation. **f** (i) Representative fluorescence heatmap of Stimulated Emission Depletion (STED) microscopy image for 5 μM SNAP-FUS-ERG mixed with 25 bp 4 × GGAA dsDNA: 0.6 μM dark dsDNA and 0.006 μM dsDNA labeled with Atto647N (100:1). (ii) Boxplot of the radial intensity distribution of dsDNA labeled with Atto647N within the hollow co-condensates. The total number N examined over

three-time in vitro droplet experiments. **g** (i) Representative fluorescence heatmap of STED image for 4 mg/mL FUS-RGG3 (FUS No. 471-504) mixed with Poly-U RNA: 20,000 ng/μL dark RNA and 100 ng/μL RNA labeled with AlexaFluor 647 (200:1). (ii) Boxplot of the radial intensity distribution of RNA labeled with AlexaFluor 647 within the hollow co-condensates. The total number N examined over three-time in vitro droplet experiments. For the boxplot in d, f(ii) and g(ii), the red bar represents median. The bottom edge of the box represents 25th percentiles, and the top is 75th percentiles. Most extreme data points are covered by the whiskers except outliers. The '+' symbol is used to represent the outliers. Statistical significance was analyzed using unpaired t test for two groups. P value: two-tailed; p value style: GP: 0.1234 (ns), 0.0332 (*), 0.0021 (**), 0.0002 (***), < 0.0001 (****). Confidence level: 95%. Scale bar: 1 μm in a(i), b(i), c; 0.5 μm in f(i); 2 μm in g(i). Source data are provided as a Source Data file.

## A mathematical model reproduces the formation of hollow co-condensates and predicts their selective capacity for DNA

To elucidate the molecular mechanism underlying hollow co-condensate formation, we developed a molecularly informed phase-field model[34]. This model incorporates three key order parameters. The first order parameter, $\eta$, represents the concentration of protein−dsDNA complexes (Open state in Fig. 3e(i)) relative to a critical threshold $\psi_C$ for phase separation. The second order parameter, $\phi$, indicates local hydrophilicity ($\phi > 0$) or hydrophobicity ($\phi < 0$) within the hollow co-condensates. Using $\eta$ and $\phi$, we derived the third-order parameter, $\chi$, which represents the dsDNA concentration within the hollow co-condensates. The free energy functional of this mesoscopic model was constructed following the classic Ohta−Kawasaki framework[35,36]. Detailed descriptions of the model development and numerical simulations are provided in the Supplementary Methods.

Figure 2a(i)−(ii) shows that dsDNA molecules containing GGAA microsatellites coat the surface of FUS-ERG condensates before slowly penetrating the interior. These observations were used to set the initial conditions of our model (Fig. 4a(i)), where the concentrations of protein and dsDNA are denoted as $\zeta_{DNA}$ and $\zeta_{protein}$, respectively. The diffusion-driven simulation, which considers only dsDNA diffusion, revealed a gradual inward migration of dsDNA molecules into the condensates (Fig. 4a(ii)), closely aligning with the experimental results shown in Fig. 2a(ii).

A central hypothesis of our model is that dsDNA, which binds to the hydrophobic ERG DBD domain (Fig. 3e(i)), preferentially localizes within hydrophobic regions of the condensate based on the Nile-red experiments (Supplementary Fig. 5b). We encoded this preference into the model's dynamics by assuming dsDNA ($\chi$) preferentially diffuses within the hydrophobic regions of the hollow co-condensates in our case (Supplementary Methods). Simulations incorporating these interactions revealed the spontaneous emergence of a stable internal cavity, defined by a sharp interface in the hydrophobicity parameter $\phi$ (Fig. 4a(iii) and Supplementary Movie 3). Additionally, the distribution of $\chi$ showed that the simulated dsDNA concentration within the shell region of the condensates (Fig. 4b) closely aligned with the STED experimental data (Fig. 3f(ii)). By varying initial values of protein and DNA concentrations, we constructed a phase diagram (Fig. 4c) that closely matches the experimental observations (Fig. 1c).

To further validate our model, we tuned two key parameters (Supplementary Methods). The first parameter, $\psi_C$, represents the critical volume fraction of the protein−dsDNA complex required for phase separation and is inversely related to protein-DNA binding affinity. In our primary simulation, setting $\psi_C$ to 0.85 yielded hollow condensates. When $\psi_C$, was increased from 0.85 to 0.95, which mimics a lower DNA-protein binding affinity, the formation of hollow condensates was not observed (Supplementary Fig. 9a and Supplementary Movie 4). Such prediction was experimentally confirmed by substituting dsDNA containing GGAA microsatellites with random dsDNA

−which exhibits lower binding affinity− also prevented the formation of hollow co-condensate, corroborating the simulation results.

The second parameter, $b_1$, quantifying the differential interaction of the protein−dsDNA complex's hydrophilic versus hydrophobic domains with the aqueous buffer, was set to 0.14 in the main simulation (Fig. 4a). When $b_1$ was decreased from 0.14 to 0.04, which indicates enhanced interaction between the hydrophobic component of protein-dsDNA complex and the working buffer compared to its hydrophilic region, the simulations revealed a failure to form hollow condensates (Supplementary Fig. 9b and Supplementary Movie 5). Notably, this result aligns with experiments using the GFP-FUS-ERG (9RA) (Fig. 1e(iv)), which exhibits increased hydrophobicity compared to wild-type FUS-ERG and fails to form hollow condensates (Supplementary Fig. 5a). This tight correspondence between model parameters and experimental perturbations substantiates our model's physical basis.

We next investigated whether our model could simulate the formation of vesicle-like hollow condensates in a distinct PRM-RNA system[17] (Fig. 3g) and sought the divergent molecular mechanisms between this system and FUS-ERG hollow condensates. It is known that RNA interacts with PRM to form a tadpole-like structure. In the RPM-RNA system oversaturated with RNA, one portion of RNA binds to PRM, generating hydrophobic regions, while the remaining RNA contributes to hydrophilic regions within the system. We therefore hypothesized that RNA ($\chi$) preferentially diffuses within both the hydrophobic and hydrophilic regions of the hollow co-condensates for such case (Supplementary Methods). Simulation with this hypothesis yielded the results shown in Fig. 4d(i)−(ii) and Supplementary Movie 6. Our simulation correctly reproduced the experimental phenotype: RNA localized to both the inner and outer surfaces of the hollow condensate (Fig. 4d-e), in contrast to the internal-surface-only localization of dsDNA in the FUS-ERG system. Such distribution closely matched the STED experimental data (Fig. 3g(ii)), revealing a distinct molecular mechanism underlying the formation of PRM-RNA hollow condensates.

Building on the ability of our mathematical model to accurately reproduce hollow co-condensate formation, we examined the outcome when FUS-ERG coexists with two dsDNA substrates differing in binding affinity. In simulations, the high- (red) and low- (blue) affinity dsDNA were assigned distinct binding affinity coefficients, and no interactions were allowed between the two dsDNA types. The results (Supplementary Methods) revealed that the low-affinity dsDNA was selectively excluded from the hollow co-condensates (Supplementary Fig. 10a). We validated this experimentally by forming hollow condensates with high-affinity GGAA-tagged dsDNA and then introducing non-specific dsDNA; as predicted, the non-specific dsDNA was excluded (Supplementary Fig. 10b, c). These findings highlight the selective nature of hollow co-condensates, driven primarily by protein−dsDNA binding affinity. This inherent property further inspired us to seek its potential applications in DNA storage systems, particularly for dynamic data manipulation.

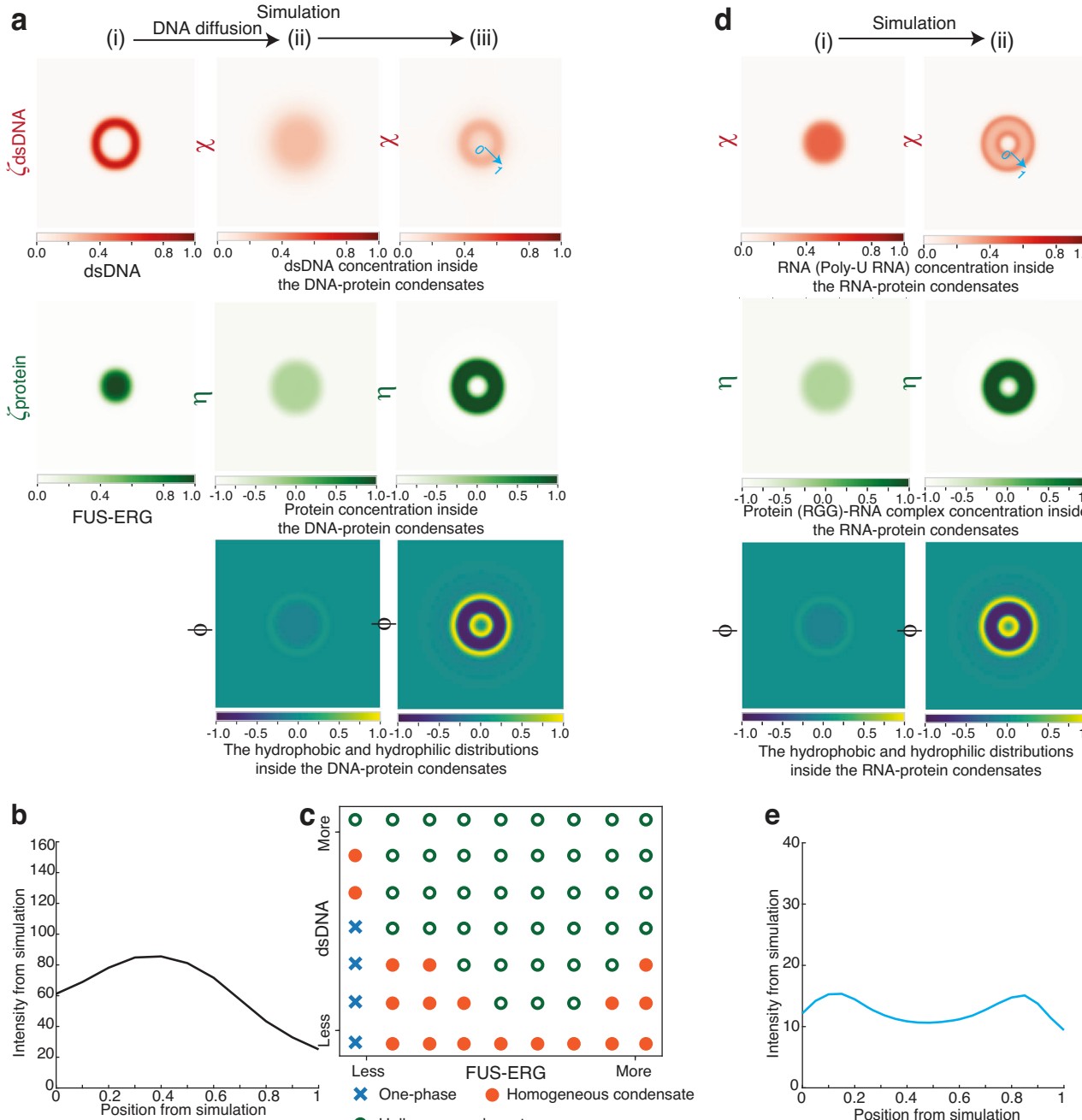

**Fig. 4 | A mesoscopic, molecularly-informed phase field model was developed to reproduce the hollow co-condensate formation. a** Simulations for FUS-ERG-DNA hollow co-condensate formation. (i) FUS-ERG molecules form biomolecular condensates with dsDNA containing GGAA microsatellites on their surface. dsDNA ($\zeta_{DNA}$) and protein ($\zeta_{protein}$) concentrations are represented. (ii) dsDNA molecules are transited into protein droplets, triggering the formation of hollow co-condensates. Order parameters denoting protein-DNA complex concentration ($\eta$), dsDNA concentration ($\chi$) and hydrophobic and hydrophilic distributions within condensates ($\phi$) are represented. (iii) Hollow architecture is observed in steady states of the model proposed. DNA is coupled with hydrophobic region of the protein-DNA complex. **b** Distributions of simulated dsDNA concentration ($\chi$) within hollow architectures. **c** States diagram of simulated structures by varying initial states in a(i). Blue cross: One-phase, which is a homogeneous state; Orange dot: homogeneous condensates; Green circle: hollow co-condensates. **d** Simulations for PRM-RNA hollow co-condensate formation. (i) RNA first forms tadpole-like diblock copolymer with PRM protein. Order parameter denoting protein-RNA complex concentration ($\eta$), RNA concentration ($\chi$) and hydrophobic and hydrophilic distributions within copolymer ($\phi$) are represented. (ii) Hollow structure is observed in simulation results. RNA is coupled with both hydrophobic and hydrophilic region of protein-RNA complex. **e** Distributions of simulated RNA concentration ($\chi$) within hollow structures. Source data are provided as a Source Data file.

## FET fusion protein-DNA hollow co-condensates enabled dynamic data manipulation for DNA-based information system

In DNA storage, data is encoded in a library of short DNA fragments each bearing a unique barcode sequence indexing the encoded content for selective random access. To examine the potential selectivity of hollow co-condensate for specific dsDNA barcodes, we first sought to determine whether hollow co-condensate formation represents a generalizable principle for FET fusion proteins with distinct DBDs and their corresponding dsDNA sequences.

To address this, we introduced a second FET fusion protein, FUS-Gal4, a model system originally developed by McKnight and colleagues[37], which we modified by incorporating the FUS RGG motif

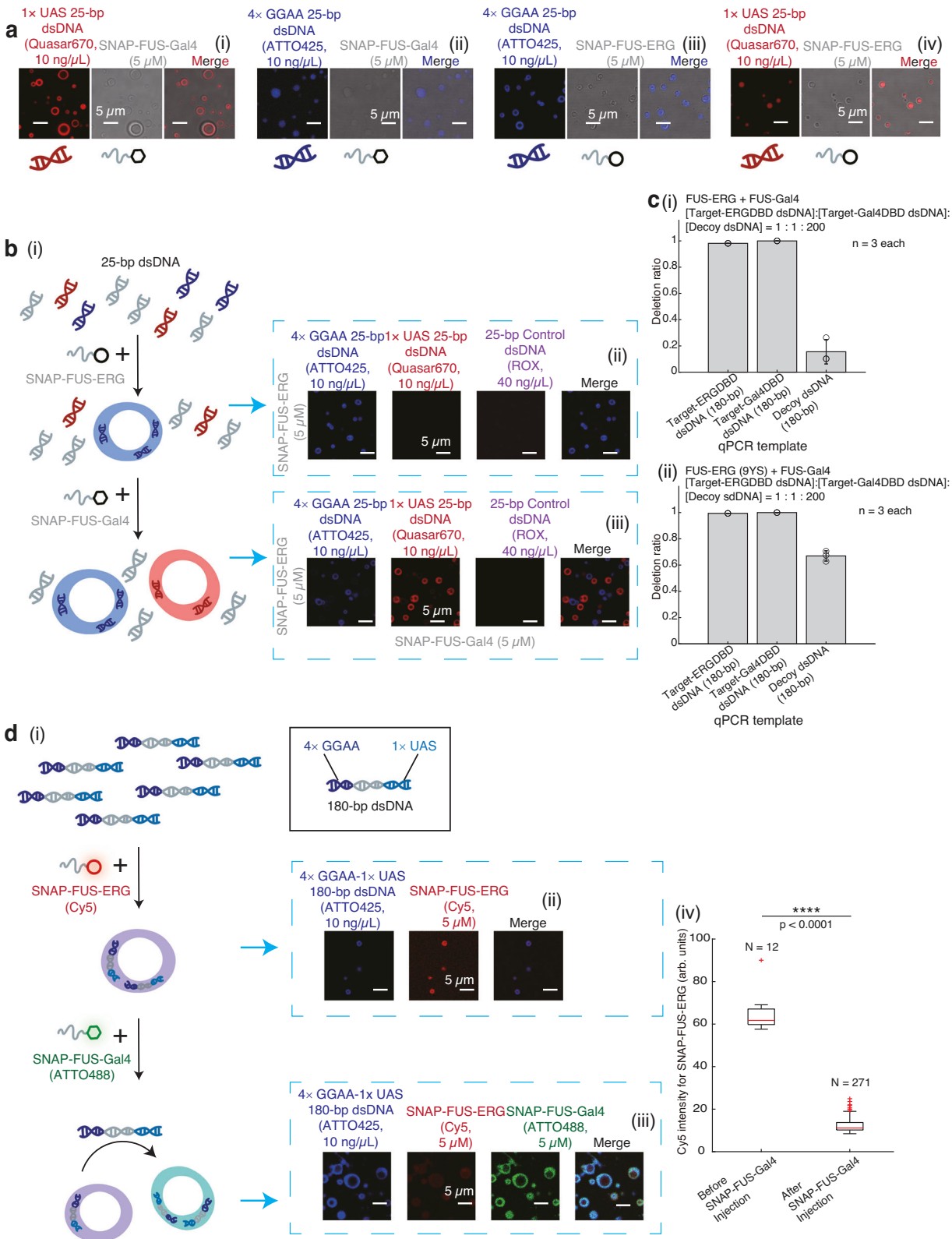

(amino acids 213–266). After purifying SNAP-FUS-Gal4 (Supplementary Fig. 11a), we observed its ability to form hollow co-condensates with dsDNA containing a single UAS sequence, the specific binding motif for the Gal4 DBD (Fig. 5a(i)). As a control, SNAP-FUS-Gal4 co-localized with dsDNA lacking the UAS sequence in homogeneous condensates (Fig. 5a(ii)). Conversely, we confirmed that SNAP-FUS-ERG forms hollow co-condensates in the presence of GGAA

microsatellites but generates homogeneous condensates when GGAA microsatellites are absent (Fig. 5a(iii)–(iv)). These findings demonstrate that the ability of FET fusion proteins to form hollow or homogeneous condensates is determined by the presence of their specific dsDNA sequences.

Leveraging the two orthogonal DBD-dsDNA interactions, we first explored whether hollow co-condensates could be utilized to sort

**Fig. 5 | These hollow co-condensates enabled dynamic data manipulation.**
**a** (i)–(iv) 5 μM SNAP-FUS-Gal4 or SNAP-FUS-ERG mixed with 10 ng/μL Quasar670-labeled 25-bp 1 × UAS dsDNA (target-Gal4DBD dsDNA) or ATTO425-labeled 25-bp 4 × GGAA dsDNA (target-ERGDBD dsDNA). **b** (i) Schematic. (ii) Fluorescence images of the addition of 5 μM SNAP-FUS-ERG into 25 bp dsDNA library: target-ERGDBD dsDNA (ATTO425), target-Gal4DBD dsDNA (Quasar670), and random dsDNA (decoy dsDNA, ROX) (1:1:4). (iii) Fluorescence images of the addition of 5 μM SNAP-FUS-Gal4. **c** Quantitative real-time PCR (qPCR) was used to evaluate the capacity of DNA selection: the addition of SNAP-FUS-ERG and SNAP-FUS-Gal4 (i) or SNAP-FUS-ERG (9YS) and SNAP-FUS-Gal4 (ii). 180-bp dsDNA library: target-ERGDBD dsDNA, target-Gal4DBD dsDNA, and decoy dsDNA (1:1:200). Independent phase separation-based DNA deletion experiments were repeated three times (*n* = 3). Error bars, mean ± s.d. **d** (i) Schematic. A 180 bp dsDNA substrate containing both 4 × GGAA and 1 × UAS sequence terminally was synthesized, with the GGAA-end labeled with ATTO425. First, 5 μM Cy5-labeled SNAP-FUS-ERG was mixed with

10 ng/μL these dsDNA substrates. Fluorescence images were shown in (ii). Second, 5 μM ATTO488-labeled SNAP-FUS-Gal4 was added, and the fluorescence images were shown in (iii). (iv) Boxplot of Cy5 intensity of SNAP-FUS-ERG in condensates of d(ii)-(iii). The total number N examined over one-time in vitro droplet experiments. For the boxplot, the red bar represents median. The bottom edge of the box represents 25th percentiles, and the top is 75th percentiles. Most extreme data points are covered by the whiskers except outliers. The '+' symbol is used to represent the outliers. Statistical significance was analyzed using unpaired t test for two groups. P value: two-tailed; p value style: GP: 0.1234 (ns), 0.0332 (*), 0.0021 (**), 0.0002 (***), < 0.0001 (****). Exact P values are as follows: Before FUS-Gal4 injection vs. after FUS-Gal4 injection, $P < 0.0001$. Confidence level: 95%. The working buffer containing 40 mM Tris-HCl (pH 7.5), 150 mM KCl, 2 mM MgCl$_2$, 1 mM DTT and 0.2 mg/mL BSA. Scale bar: 5 μm in a, b(ii)-(iii), and d(ii)-(iii). Source data are provided as a Source Data file.

---

specific dsDNA substrates from a dsDNA library. We mixed three dsDNA substrates in a 1:1:4 ratio of 25-bp 4 × GGAA dsDNA (target-ERGDBD dsDNA), 25 bp dsDNA containing a 1 × UAS sequence (target-Gal4DBD dsDNA), and 25 bp random dsDNA (decoy dsDNA, Fig. 5b(i)). Initially, we added SNAP-FUS-ERG into the dsDNA library to observe blue-colored hollow co-condensates (Fig. 5b(ii)), affirming that SNAP-FUS-ERG selectively binds to target-ERGDBD dsDNA, while target-Gal4DBD dsDNA and decoy dsDNA remain in the solvent phase. Building upon this observation, when we subsequently introduced SNAP-FUS-Gal4, a secondary set of hollow co-condensates emerged exhibiting a red hue (Fig. 5b(iii)), indicating the spatial sorting of dsDNA into separate condensates according to their distinct barcodes.

We further demonstrate through control experiments that the hollow co-condensate architecture is essential for selective dsDNA sorting. When GFP-FUS-ERG (9YS) was mixed with target-ERGDBD dsDNA or target-Gal4DBD dsDNA (used as control dsDNA), no hollow co-condensates formed (Supplementary Fig. 12a and Fig. 1e(iv)). Repeating the experiment shown in Fig. 5b, but substituting the initial addition with GFP-FUS-ERG (9YS), revealed that this variant not only failed to selectively bind to target-ERGDBD dsDNA (Supplementary Fig. 12b(i)), but also disrupted the subsequent sorting of target-Gal4DBD dsDNA by SNAP-FUS-Gal4 (Supplementary Fig. 12b(ii)). Additionally, the process significantly affected decoy dsDNA substrates, ultimately resulting in the complete failure of DNA sorting.

These observations strongly imply that hollow co-condensates could be harnessed for targeted DNA deletion within DNA libraries. Crucially, we also examined whether the non-specific dsDNA substrates in the library remained unaffected during this process. To evaluate selective DNA deletion efficiency, we prepared a dsDNA library containing target-ERGDBD dsDNA, target-Gal4DBD dsDNA, and decoy dsDNA at a molar ratio of 1:1:200. We independently used GFP-FUS-ERG and SNAP-FUS-Gal4 to sequester and pull-down target-ERGDBD and target-Gal4DBD dsDNA, respectively, by iterative centrifugation (Methods). After four iterations, quantitative real-time PCR (qPCR) revealed that target-ERGDBD dsDNA and target-Gal4DBD dsDNA was removed from the library by pulling down with specific FET fusion proteins hollow condensates with an efficiency of 98.1 ± 0.1% and 100 ± 0.0%, respectively, while the decoy dsDNA was removed at a much lower rate of 15.6 ± 9.3% (Fig. 5c(i)) according to the calibration of each dsDNA in the library (Supplementary Fig. 13). As a control, we repeated the experiment using the mutant GFP-FUS-ERG (9YS) instead of the wild-type GFP-FUS-ERG. Although target-ERGDBD dsDNA and target-Gal4DBD dsDNA were sorted and removed at similar efficiencies of 99.4 ± 0.0% and 100 ± 0.0%, respectively, decoy dsDNA was strongly co-depleted by 67.0 ± 3.9%, a 4.3-fold excess removal compared to deletion by the wildtype FUS-ERG (Fig. 5c(ii)).

Lastly, we challenged the hollow co-condensate system for dynamic and hierarchical data selection. More specifically, we implemented a two-step sequential sorting logic based on the barcode-

protein binding specificity (Fig. 5d(i)). A 180-bp dsDNA substrate containing both 4 × GGAA and 1 × UAS sequence terminally was synthesized, with the GGAA-end labeled with ATTO425. Initially, we mixed the dsDNA substrate with SNAP-FUS-ERG and observed hollow co-condensates (Fig. 5d(ii)), indicating the binding of SNAP-FUS-ERG to the 4 × GGAA sequence. Afterwards, SNAP-FUS-Gal4 was introduced into the sample. We found that SNAP-FUS-Gal4 displaced SNAP-FUS-ERG from the condensates, forming new hollow co-condensates with the dsDNA substrates (Fig. 5d(iii)). This outcome suggests that SNAP-FUS-Gal4 effectively competes with SNAP-FUS-ERG for binding to the dsDNA substrates and as a result, the FUS-ERG condensate disassembled (Fig. 5d(iv)). In stark contrast, with the order of addition switched to SNAP-FUS-Gal4 in the first step followed by SNAP-FUS-ERG in the second, all components co-localized into homogeneous condensates (Supplementary Fig. 14a). Considering the stronger binding affinity of Gal4 DBD to UAS sequence compared to ERG DBD to GGAA microsatellites (Supplementary Fig. 11a(iii) and 1b(iii)), the sequential logic of data manipulation necessitates the strict temporal programming of the hollow co-condensate system.

As control experiments, we evaluated several variant FET fusion proteins, including FUS-Gal4 lacking the RGG motif (GFP-FUS-Gal4 (no RGG), Supplementary Fig. 11b), GFP-FUS-ERG (9YS) (Supplementary Fig. 1c), and GFP-FUS-ERG (9RA) (Supplementary Fig. 1f). Independent assessments confirmed that these variants were unable to drive hollow co-condensate formation, rather, they formed homogeneous condensates with target dsDNA (Supplementary Fig. 14b, d(i), and e(i) & Fig. 1d(ii) and e(iv)). The experiment depicted in Fig. 5d was repeated, substituting the wildtype fusion proteins with their mutants (Supplementary Fig. 14c-e). Despite maintaining the correct order of protein addition, target selection failed at the step where the mutant proteins were introduced. These findings underscore the critical requirement for both FET fusion proteins to possess the intrinsic capability to form hollow co-condensates in order to achieve the outcomes observed in Fig. 5d.

## Discussion

In this study, we demonstrate that the FET family fusion oncoprotein FUS-ERG forms hollow co-condensates with dsDNA containing GGAA microsatellites. Using biochemical assays, super-resolution imaging, and mathematical modeling, we revealed the formation of hollow condensates through the process of nested asymmetric phase separation. Moreover, we show that the self-organization and morphology control of FUS-ERG hollow co-condensates can be leveraged for DNA-based information manipulation, enabling precise DNA deletion within dsDNA libraries and dynamic, hierarchical data selection.

The hollow condensates observed in Fig. 1 are consistent with the Ostwald rule of stages, suggesting that these structures may represent metastable intermediates along the pathway toward more thermodynamically stable states. This metastability appears tunable,

in part through homotypic interactions mediated by physical cross-links between sticker residues, such as tyrosine–arginine contacts (Fig. 1d, e). The finding that dsDNA containing GGAA microsatellites promotes hollow condensate formation suggests that the DBD can reveal and stabilize an intrinsic propensity for such architectures[31], either through site-specific recognition (Fig. 2a) or surface-mediated interactions (Supplementary Fig. 2c). These observations are consistent with previous reports of vesicle-like structures formed by FUS proteins under subsaturated conditions[38], supporting the existence of a latent vesicular phase. We further speculate that these hollow condensates may correspond to a Lifshitz point or fall along a Lifshitz line, with DNA acting as a trigger to unmask this phase behavior. This possibility opens new avenues for probing the thermodynamic and kinetic landscape governing condensate morphology.

Biomolecular condensates can be modeled using three primary approaches: microscopic, mesoscopic, and macroscopic models. Microscopic models, such as molecular dynamics and coarse-grained simulations, provide detailed insights into the structural and dynamic properties of condensate formation. This approach has been successfully applied to hollow co-condensates formed by PRM and RNA[17]. However, their high computational demands make it impractical for interrogating more complex condensate architectures such as ours. Macroscopic models, including phase-field models based on Flory-Huggins theory[16,39], effectively capture component interactions and macroscopic phase structures within condensates. Despite these strengths, macroscopic models fall short in resolving local, sub-compartmental characteristics, such as hydrophobicity distributions and fine-scale component organization within the shell of hollow co-condensates —features that are critical for our study. To overcome these limitations, we adopted a molecularly informed phase-field model[34], which operates at the mesoscopic scale. This model incorporates order parameters $\phi$ and $\chi$ (Fig. 4), enabling a seamless integration of macroscopic phase structures with local molecular characteristics. Compared to microscopic and macroscopic approaches, the model offers a balanced and computationally feasible strategy to explore the complex behavior of hollow co-condensates in our system (Supplementary Methods).

Despite our observation of dsDNA containing GGAA microsatellites translocating into condensates (Fig. 2), the molecular mechanism underlying this translocation remains elusive. To address this, we propose an interesting hypothesis: Initially, Closed-state proteins form biomolecular condensates with the working buffer, forming a condensed phase of proteins within homogeneous condensates. Upon binding of dsDNA substrates containing GGAA motifs, conformational changes are induced, resulting in the formation of Open-state protein–dsDNA complexes. Due to their relative scarcity, these Open-state complexes likely establish a loosened phase[40], segregating from the condensed phase of Closed-state proteins and creating microscopic channels within the condensates. As additional dsDNA substrates bind to FUS-ERG molecules on the condensate surface, these channels facilitate the diffusion of Open-state complexes into the condensate interior. This process is likely driven by a combination of interfacial depletion effects[41,42] and hydrophobic interactions. As the Open-state complexes diffuse inward, they displace Closed state proteins, establishing an inner interface enriched with Open-state complexes. Collectively, this hypothesis suggests that hydrophobic interactions and interfacial depletion effects synergistically drive the internal phase separation of Open-state proteins, ultimately leading to the formation of diffusion-controlled hollow co-condensates. Further investigation into this proposed mechanism represents an exciting avenue for future research.

Given the prevalence of multicomponent phase-separated cellular bodies in living cells, deciphering the molecular mechanisms of condensate architectures has important biological relevance. For instance, in mammalian nucleoli, dense fibrillar components (DFCs) are one type of biomolecular condensates, which envelop another type of biomolecular condensates—fibrillar centers (FCs), creating "Russian doll" structures that facilitate efficient transcription at the FC/DFC interface[43]. Secondly, in the in vitro reconstitution of mitochondrial transcription machinery, the formation of hollow co-condensates was shown to play a crucial role in regulating transcription rates[18]. The focus of our study, the FET family fusion oncoproteins, have been found to form biomolecular condensates at genomic binding sites[44,45], and these condensates can recruit RNA polymerase II to induce excessive gene transcription, causing sarcomas and leukemia[46,47]. Whether the condensate architecture impacts transcription regulation is one interesting topic for the future study.

Finally, the unique molecular mechanism of hollow co-condensates provides a novel methodology in DNA-based information manipulation. Although DNA self-assembly has been successfully applied to the field of DNA-based data storage[48–50], we provide the first direct demonstration of using protein-DNA self-assembly for spatial information manipulation. By reconstituting hollow condensates in synthetic DNA storage systems, we showed that this specific architecture is necessary for highly selective and highly dynamic data manipulation.

For the targeted DNA deletion within DNA libraries (Fig. 5b), we compared the separation efficiency of our method with previous methods. So far, data sorting has been demonstrated by using complementary DNA probes combined with solid-state purification or site-specific restriction enzyme cleavage of a particular DNA from an adsorption matrix[48,51]. In particular, primer-based magnetic separation was shown to have <75% efficiency with a 20-nt primer length and optimal temperature[48]. Restriction enzyme cleavage was shown to have around 40% of target-strand retention[52]. Our phase separation-based manipulation, in comparison, achieved almost 100% removal while affecting only 15% of decoy DNA in an oligo system of notably lower signal-to-noise ratio (1:1:200, Fig. 5c). Spatial regulation by protein-DNA assembly, compared to widely used DNA self-assembly approaches, could leverage several unique properties for the observed high sorting efficiency and accuracy. First, complex protein-DNA interactions offer enhanced binding efficiencies as well as broader and tunable affinity profiles. Second, the condensate morphology induced by protein conformational switch and the heterogeneity of internal DNA distribution further enhanced the condensate's target selectivity (Supplementary Fig. 10a).

On the other hand, the results on hierarchical data sorting (Fig. 5d) suggested that the hollow co-condensates are also sufficiently dynamic for compositional exchange. Therefore, the architecture acts as a temporary storage depot for DNA molecules, allowing for sequential manipulations of data encoded in DNA, while ensuring a high indexing specificity to the programmed temporal logic. Therefore, our phase separation-based hollow co-condensate system represents a new vessel for in-storage dynamic regulation and data processing in memory. Building upon this basic principle, it is possible to include more dsDNA-DBD binding for finer information sorting, to employ more complex architectures for precise and flexible multi-component control, and to enable sophisticated information manipulation through protein engineering and protein-based regulation.

In summary, these findings provide important insights into the biophysical mechanisms underlying multicomponent phase-separated cellular bodies, and also offer innovative strategies for manipulating DNA-based information.

## Methods

### Protein expression and purification

A list of the proteins' sequences used in this study is shown in Supplementary Information. sfGFP or SNAP tag were fused at the N-terminus of the FET fusion proteins used in this study with a 4 × GGS

linker. The plasmid vector was a gift from the Pilong Li lab (Tsinghua University, China).

Proteins were expressed and purified using affinity chromatography as described in our earlier work[45]. Specifically, plasmids were transformed into *E. coli* strain BL21 (DE3) (Vazyme, Cat# C504-02), and then cultured overnight on LB plates. Monoclonal strains were inoculated into 10 mL of liquid LB medium and cultured overnight at 37 °C with shaking at 220 rpm. This culture was then inoculated into 2 L of liquid LB medium, followed by culturing at 37 °C with shaking at 220 rpm until the $OD_{600}$ reached 0.8. 0.5 mM IPTG was added, and the culture was incubated at 16 °C with shaking at 180 rpm for 18 h to induce protein expression.

The bacterial culture was centrifuged at 4 °C and 4000 × g for 15 min. The pellet was resuspended in lysis buffer (50 mM Tris-HCl (pH 7.4), 1 M KCl, 1 M Urea, 10 mM Imidazole, 1.5 mM β-Mercaptoethanol (βME), and 5% Glycerol), followed by sonication. After centrifugation at 4 °C and 18,000 × g for 30 min, the supernatant was collected and loaded onto a Ni-NTA resin (Thermo Scientific, Cat# 88221), which can bind to the target proteins containing 6 × His-tag via $Ni^{2+}$ ions. Once the resin was washed using wash buffer (50 mM Tris-HCl (pH 7.4), 1 M KCl, 1 M Urea, 50 mM Imidazole, 1.5 mM βME, and 5% Glycerol), the target protein was eluted with elution buffer (50 mM Tris-HCl (pH 7.4), 1 M KCl, 1 M Urea, 500 mM Imidazole, 1.5 mM βME, and 5% Glycerol) and stored in lysis buffer at −80 °C after further purification by gel filtration with a Superdex 200 column (GE Healthcare).

### Fluorescence labeling

Part of SNAP-FUS-Gal4 and SNAP-FUS-ERG were labeled by Atto488-NHS ester (Sigma aldrich, Cat#72464) and Sulfo-Cyanin5 NHS-Ester (Lumiprobe, Cat#23320) respectively. Proteins were mixed with the dye in the carbonate buffer at room temperature for 1 h, and then the buffer was exchanged to storage buffer (including 20 mM Tris-HCl (pH 7.5), 1 M KCl, 1 M Urea, 5% glycerol, 1 mM β-mercaptoethanol), on the meanwhile, free dyes were removed through centrifuge using 0.5 mL Centrifugal filters 30 kDa (Amicon). The labeling efficiency for all samples was observed to be ≥80% (UV–Vis absorption measurements).

### Oligo Preparation

1 × UAS binding sites and microsatellite sequence (25 × ), 4 × , 2 × GGAA repeats and control sequence dsDNA were directly ordered from Ruibiotech. dsDNA labeled by fluorescence dyes were synthesized by Sangon Biotech.

DNA longer than 25 bp for the in vitro droplet experiment was prepared by PCR from the target vectors. Those PCR products were then purified by the spin column (Tiangen).

25-bp dsDNA substrates used for the electrophoretic mobility shift (EMSA) assay and droplet assays were generated by a slow annealing protocol. In the annealing system, 1 μM fluorescent-labeled top strand and 1.2 μM dark bottom strand were added in an annealing buffer containing 40 mM Tris-HCl (pH 8.0), 50 mM NaCl, and 10 mM $MgCl_2$. Then we loaded the 1.5 ml Eppendorf tube in a 1 L beaker of 800 ml water, heated it to 95 °C for 5 min, and put it on a wood bench for a 3 h slow cooling to room temperature. The finally obtained fluorescent dsDNA was 1 μM.

FUS RGG3 (Synthesized by Sangon Biotech) and polyU RNA (Sigma-Aldrich, Cat#P9528) were resolved in the buffer containing 10 mM Tris-HCl (pH 7.5) and DEPC $ddH_2O$ at the concentration of 20 and 50 mg/mL, respectively.

### In vitro droplet experiment and data analysis

**Samples preparation.** Frozen protein aliquots were thawed and kept on ice. Proteins were centrifuged at 4 °C, 13,800 × g to separate the aggregations. Concentrated protein solutions at high salt (1 M KCl) were diluted into low-salt buffer to reach final buffer conditions of 150 mM KCl. To create the final solutions, the reagents were added to an Eppendorf tube in the following order: protein and dsDNA as indicated in the figures, then added working buffer (40 mM Tris-HCl (pH 7.5), 150 mM KCl, 2 mM $MgCl_2$, 1 mM DTT and 0.2 mg/mL BSA) to achieve a final volume of 10 μL. Solutions were pipetted up and down to ensure complete mixing. Approximately 10 μL of solution was added to 384 well glass bottom plate (Cellvis, Cat.# P384-1.5H-N) and incubated for ~30 min at room temperature prior to imaging.

**Nile red staining.** Nile red staining was used to quantify the hydrophobicity of the condensate. After the droplet formed on the 384 well glass bottom plate, Nile red staining solution (ABmole, Cat.# M5118) co-incubated with droplets for 20 min at room temperature. All pictures were taken with a Leica microscope. Ex/Em = 552/636 nm.

**Condensate diameter measurement.** The diameter of condensates was measured in ImageJ (Version 1.53f51). Before particle analysis, the background of the image was corrected using the menu command *Subtract background*. The radius was set to 50.0 pixels. Then, an auto threshold range was set to tell the droplets of interest apart from the background. To analyze the droplets, we used the menu command Analyze particles. In order to exclude the noise in the image, we set the minimum particle size to 0.5 μm² and the range of the circularity was 0.8–1.0. For shell-like images, "Include holes" was chosen to get the whole area of the droplet. The condensate diameter was calculated from the area data.

### Fluorescence recovery after photobleaching (FRAP)

The samples were prepared as described and the mixed samples were incubated at room temperature for 30 minutes. FRAP experiments were performed by a spinning-disk confocal microscope (UltraView VoX) to bleach an area of 1.0 × 1.0 μm inside each droplet. Laser power was adjusted to 100% for photobleaching. The first frame was immediately taken after bleaching, following by a chronological series of photos with time interval of 15 s. For all FRAP experiments, the time point for photobleaching was defined as 0 second. Data of each bleached spots were processed by the software Volocity 6.3.5, and the normalized intensity was used to plot the recovery curve.

### Super-resolution imaging of the hollow co-condensates

For STED sample preparation, in the FUS-ERG shell-like structure observation, 25 bp dsDNA containing 4 × GGAA was labeled with Atto647N, and then mixed with non-labeled 25 bp dsDNA with the same sequence in a ratio of 1:100 to the final concentration of 10 ng/μl. dsDNA and 5 μM SNAP-FUS-ERG were incubated in the working buffer containing 40 mM Tris-HCl (pH 7.5), 150 mM KCl, 2 mM $MgCl_2$, 1 mM DTT and 0.2 mg/mL BSA on the glass bottom dish (Cellvis, Cat.# D35-10-1-N). For RGG-polyU hollow co-condensate observation, Alexa-fluor 647 labeled 50-nt polyU was mixed with non-labeled polyU in a ratio of 1:200 to the final concentration of 20 mg/mL. PolyU RNAs were incubated with 4 mg/mL FUS-RGG3 in the same working buffer on the glass bottom dish for 30 min before imaging. For imaging, the samples were recorded using STEDYCON STED microscopy (Abberior Instruments GmbH, Göttingen, Germany) equipped with a CFI Plan Apochromat Lambda D 100 × oil, NA1.45 objective (Nikon, Tokyo, Japan). Pixel sizes of 20–30 nm were used for STED nanoscopy. Droplets stained with Atto647N labeled dsDNA or Alexa-fluor 647 labeled single-stranded RNA was excited at 561 nm wavelength and STED was performed using a pulsed depletion laser at 775 nm wavelength with gating of 1–7 ns and dwell times of 10 μs. STED was performed at 775 nm wavelength with gating set to 1–7 ns. Dwell times of 10 μs were used. Alexa 488 was excited at 485 nm and recorded in the confocal mode. The fluorescence signal was usually accumulated over 5–10-line steps.

### STED data analysis

Analysis of STED data analysis consists of three main steps: Enhancement, Annotation and Measurements. The initial step focuses on

enhancing the quality of the microscopy images to facilitate a clearer understanding of the structures being examined. This is achieved through a process that begins with the application of a discrete Fourier transformation $\hat{x} = F(x)$ to the input image x, effectively translating it into the Fourier K-space. This transformation is pivotal as it allows for the assumption that white noise, which typically manifests as high-frequency components in the spatial domain, corresponds to constants in the K-space. To mitigate the impact of this noise, a low-pass filter is applied to eliminate all frequency components above a pre-determined threshold, typically set at half the grid length. By zeroing out these high-frequency components, the process effectively removes the noise when the image is transformed back into the spatial domain through an inverse Fourier transformation. This step is crucial for enhancing the clarity and readability of the image, paving the way for more accurate annotation and measurement. The next phase, annotation, involves the meticulous marking of the external and internal surfaces of the condensates. Utilizing annotation tools such as CVAT, surfaces are denoted as ellipses, with the internal surface selected manually based on an automatic calibration of the external surface. This step is fundamental in defining the regions of interest for subsequent analysis, ensuring that measurements are conducted with precision. The final step in the analysis process is the measurement of relative signal strength within the condensates. By generating 20 intermediate ellipses through linear interpolation between the annotated internal and external surfaces, researchers can systematically count the area-normalized signals within each segmented region. This meticulous approach allows for the quantification of signal strength across the condensate, from the internal to the external surface. The data is then plotted with the normalized distance from the internal surface on the x-axis, providing a comprehensive visualization of the signal distribution within the condensate.

### DNA deletion assay

In this assay, the DNA library contained three different sequences in same length, one has a $4 \times$ GGAA adapter at the 5' of the dsDNA, one has a $1 \times$ UAS adapter at the 5' of the dsDNA while another does not. $5 \mu M$ SNAP-FUS-ERG and $5 \mu M$ SNAP-FUS-Gal4 were mixed with the DNA library in the working buffer containing 40 mM Tris-HCl (pH 7.5), 150 mM KCl, 2 mM MgCl$_2$, 1 mM DTT and 0.2 mg/mL BSA to a total volume of $20 \mu L$. The ratio of dsDNA with target adapter was 1:200, which means the amount of both dsDNA with $4 \times$ GGAA and $1 \times$ UAS was 5 ng, and control dsDNA was 1000 ng. The mixture was incubated at room temperature for 30 minutes in a $200 \mu L$ tube. Then centrifuge it at $13,800 \times g$ under room temperature for 15 minutes to separate the dense phase containing SNAP-FUS-ERG and dsDNA with GGAA adapter. Move $15 \mu L$ supernatant carefully to a new $200 \mu L$ tube and supplement $5 \mu L$ 2.5 $\mu M$ SNAP-FUS-ERG and 2.5 $\mu M$ SNAP-FUS-Gal4 into the system. Repeated those operations for four times, added $5 \mu L$ working buffer into the supernatant after last centrifugation. Most of the dsDNA with GGAA/UAS adapter were supposed to be removed with the dense phase after the treatments. These sequence-specific condensates sediment upon centrifugation, resulting in a reduced concentration of the corresponding DNA in the supernatant. To quantify this effect, we performed qPCR analysis (as described in qPCR quantification) to measure the remaining amounts of specific dsDNA sequences.

### qPCR quantification

In order to quantify the deletion efficiency accurately, we used qPCR to indicate the remaining dsDNA concentration. Firstly, we mixed the three dsDNA together in the ratio of 1:1:200 as mentioned before, without proteins. Diluted each of the dsDNA in the mixture to 0.0005 ng/$\mu L$ separately. To make the titration curve, the diluted templates of the dsDNA were added 0.5, 1, 1.5, 2L, and 2.5 $\mu L$, respectively, to 10 $\mu L$ to make five dilutions. dsDNA was further diluted 2 folds

in the final qPCR SYBR mix, so the final concentration in the qPCR was $1.25 - 6.25 \times 10^{-5}$ ng/$\mu L$. As the initial concentration of the two different sequences dsDNA before deletion was the same as that in the titration curve, so the titration folds were consistent with the control system. Afterwards, prepared the qPCR master mix, which contained 10 $\mu L$ $2 \times$ Taq Pro Universal SYBR qPCR Master Mix (Vazyme, Cat.Q712), 0.4 $\mu L$ forward and reversed primers (10 $\mu M$). The master mix was added into the diluted template sample and pipetted to mix well. Using the following thermal profile:

| Procedure | Temperature | Time |
|---|---|---|
| Hot start | 95 °C | 3 min |
| 40 cycles | 95 °C | 3 s |
| | 65 °C | 10 s |
| Melting curve | According to the default program | |

To quantify the fold change of the dsDNA with or without GGAA adapter according to the qPCR results, we generated the standard curve from the control template (dsDNA mix without protein treatment) by plotting the log dsDNA concentration against the Ct values, and the curves were fitted linearly (Supplementary Fig. 13). The concentration of dsDNA with or without GGAA adapter were determined by the Ct value according to the standard curve. The calibration method was the same for dsDNA with a $1 \times$ UAS adapter, except for the dilution fold of the qPCR template. The "deletion ratio" is calculated as the proportion of a given dsDNA sequence that remains in the supernatant after centrifugation, relative to its original concentration in the dsDNA library before condensate formation and centrifugation. A high deletion ratio, therefore, indicates efficient removal/depletion of the target sequence from the library—i.e., that it was selectively incorporated into the sedimented condensates.

### Electrophoretic mobility shift assay (EMSA)

EMSA experiments (Supplementary Fig. 1 and 9) were performed to test the binding affinity of proteins-DNA. For Gal4 related proteins, dsDNA with/without a Gal4 target site (17-bp) was used (25 bp, 0.1 $\mu M$). The Gal4DBD binding site was in the middle, connecting with a random 4-bp sequence on either side. For FUS-ERG related proteins, dsDNA with GGAA microsatellites was used (306 bp, 0.1 $\mu M$). All dsDNA fragments in Gal4 and FLI1 EMSA were labeled with Alexa-fluor647 at the 5' end. The working buffer included 40 mM Tris-HCl (pH 7.5), 150 mM KCl, 2 mM MgCl$_2$, 1 mM DTT and 0.2 mg/mL BSA.

The samples with different molar ratio of [protein]: [substrate] were pre-incubated for 30 minutes at room temperature, and then were loaded on a 8% native polyacrylamide gel electrophoresis (PAGE) gel. The size of the gel was 1.5 mm-thick and 20 cm-long. The gel was run in $1 \times$ TBE buffer (0.1 M Tris-base, 0.1 M Boric acid and 2 mM EDTA) under 100 V voltage for 45 minutes. The protocol of 8% native PAGE gel (10 mL) was: (1) 2.67 mL Acrylamide/bis (30% 29:1; Meilunbio, Cat# MA0071); (2) 2 mL $5 \times$ Tris/borate/EDTA (TBE) electrophoresis buffer; (3) 166 $\mu L$ Ammonium persulfate (APS, 10%); (4) 8 $\mu L$ TEMED (Amresco, Cat# 0761-100 ML); (5) Replenish to 10 mL with distilled H$_2$O. The TBE PAGE gel for RNA substrate contained 4.5% Acrylamide/bis.

For FLI1 related proteins, the dsDNA substrates were incubated with indicated concentrations of proteins in the working buffer for 30 minutes under room temperature. 1.2% agarose gel was used. The gel was run in $1 \times$ TBE buffer under 120 V voltage for 30 min. The protocol of 1.2% agarose gel was: dissolving 0.36 g agarose powder (Biowest, Cat# BY-R0100) in 30 mL $1 \times$ TBE buffer and cooling it down until it became solid gel. All the EMSA results were acquired by GE Amersham typhoon detected by Cy5 filter channel.

## Cell culture

Human cancer cell line U2OS was a general gift from Yihan Lin lab (Peking University, China). In the cell line, Dulbecco's modified Eagle's medium (Gibco DMEM, Thermo Fisher, Cat# 11965092) with 10% fetal bovine serum (FBS, Gibco, USA, Cat# 10099141) was used as a culture medium under 5% $CO_2$ at 37 °C.

## Cell transfection and plasmid construction

All the overexpression cDNA, including FUS-DDIT3-mEGFP-overexpressed and FUS-DDIT3 9RA-mEGFP-overexpressed were cloned into L379 vector respectively. And the plasmid was transfected into cancer cells using Lipofectamine (Meilunbio, China, MA0672) at 1:1000. In brief, DNA dilution was prepared by mixing Opti-MEM Serum Medium (Gibco, USA, Cat# 31985062), relative cDNA and Lipofectamine. After 20 min of quiescence at room temperature, the mixture was added to the target cells. After prepared by mixing Opti-MEM Serum Medium (Gibco, USA) and Lipofectamine, the mixture was added to the U2OS cells. Then, removed the medium after 6 h infection and cultured the cells with Opti-MEM Serum Medium for 20 h. After transfection for 20 h, the living cell images were captured directly. For DNA observation, cells were fixed by 4% PFA, and incubated with Hoechst 33258 (Solarbio Life Science, Cat# IH0060) for 10 minutes. All Images were captured by Leica SP8 confocal microscope.

## Unpaired t test

Statistical significance was evaluated based on Student's t-tests (Prism 9 for macOS, Version 9.1.0 (216), March 15, 2021, GraphPad Software, Inc.). Test was chosen as unpaired t test. P value style: GP: 0.1234 (ns), 0.0332 (*), 0.0021 (**), 0.0002 (***), < 0.0001 (****).

## Box-plot

The function of "boxplot" in MATLAB software (R2015a, 64-bit, February 12, 2015) was used to plot the boxplots in Figs. 2, 3, 5, and Supplementary Fig. 5. For each boxplot, the red bar represents median. The bottom edge of the box represents 25th percentiles, and the top is 75th percentiles. Most extreme data points are covered by the whiskers except outliers. The '+' symbol is used to represent the outliers.

## Statistics and reproducibility

No statistical method was performed to predetermine sample sizes. No data were excluded in this study. For the high-throughput microscopy experiments, the sample size, statistical test used, and the detail of the analysis are provided in figure, extended data figure captions, and Methods. All in vitro droplet assays in Figs. 1a–e, 2a, b, 3a–c, and 5a, b, d were repeated three times. Supplementary Fig. 1b(iv), 2a, c, 8, 10b, c, 12, and 14 were repeated three times. Figures show one representative experiment.

## Reporting summary

Further information on research design is available in the Nature Portfolio Reporting Summary linked to this article.

## Data availability

All data generated or analyzed during this study are included in this published article (and its supplementary information files). Source data are provided with this paper.

## Code availability

The code repositories used to develop the model and perform the STED analysis are publicly available and have been deposited in [https://github.com/michaelGuo1204/FETshell_STED][53] and [https://github.com/michaelGuo1204/FETshell_PFModel][54] under MIT license. The specific version of the codes associated with this publication are archived in Zenodo and are accessible via https://doi.org/10.5281/ zenodo.17132503 (Main Model) and https://doi.org/10.5281/zenodo.17132510 (STED analysis) respectively.

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

## Acknowledgements

We thank the Peking Nanofab for process support. We thank the contributions of the Engineering Research Center for Semiconductor Integrated Technology, Institute of Semiconductors, Chinese Academy of Sciences. We thank Dr. Jun Cheng for assisting us for protein purification and in vivo experiments. We thank Yijia Bian (Undergraduate in the Integrated Science Program, Peking University) for cartoon figures plotting. We thank the Pilong Li laboratory for their generous support of this work, particularly for providing the plasmid vector and for their valuable comments on the manuscript. We thank Dr. Luhua Lai, Dr. Chun Tang, Dr. Yujie Sun, and Dr. Zhiyuan Li (Peking University), Dr. Wei Ji (Institute of Biophysics, Chinese Academy of Sciences), Dr. Xueping Zhao (University of Nottingham Ningbo China), Dr. Xiangze Zeng (Hong kong Baptist University), and the members of the Z.Q. laboratory for comments on the manuscript. This work was supported by the National Key Research and Development Program of China (2023YFF1205600 to Z.Q). This work was supported by the National Natural Science Foundation of China (Grant No. T2225009 (Z.Q.)). This work was supported by the National Key Research and Development Program of China (2024YFA0919500 to L.Z. and 2023YFF1206100 to L.Q.). This work was supported by the National Natural Science Foundation of China (Grant No. T2321001, T2222022 (Y.X.), 32088101, 12225102 (L.Z.), 12288101 (L.Z.), 12426653 (L.Z.), and 12090054 (L.Q.)).

## Author contributions

L.Z. prepared biological samples, designed and conducted all experiments, performed data analysis, and wrote the manuscript. Q.G. and L.Z. built up the mathematical model, performed simulation and theoretical analysis, and wrote the manuscript. C.L. assisted L.Z. and performed all experiments for the manuscript revision. K.Z. and Z.C. conducted the super-resolution imaging experiments. Y.C. assisted C.L. in conducting some experiments for the manuscript revision. B.J. assisted L.Z. with the qPCR experiments. Y.X. supervised the project, experimental designs, and wrote the manuscript. L.Q. designed the experiments for DNA manipulation and wrote the manuscript. Z.Q. supervised the project, experimental designs, and data analysis, and wrote the manuscript with input from all authors.

## Competing interests

The authors declare no competing interests.
