## [Transparent Peer Review file · Nature Communications]

Deciphering the molecular mechanisms of FET fusion oncoprotein–DNA hollow co-condensates

Corresponding Author: Professor Zhi Qi

Version 0:

Reviewer comments:

Reviewer #1

(Remarks to the Author)

Deciphering the molecular mechanisms of FET fusion oncoprotein–DNA hollow co-condensates by Linyu Zuo et al. describes the combination of biochemical assays, super resolution imaging and mathematical modeling to understand the phase behavior of hollow co-condensates describing nested asymmetric phase separation in FUS-ERG (a fusion oncoprotein found in prostate cancer and Ewing sarcoma) and 25bp 4xGGAA dsDNA complexes which bind specifically to the DNA binding domain of ERG. The authors show that this morphology enhances data sorting enabling targeted DNA deletion in dsDNA libraries and hierarchical DNA selection.

Initially the liquid liquid phase separation morphology of FUS-ERG GFP fusions is studied by itself and in complex with fluorescently labelled 25bp 4xGGAA (specific to the DNA binding domain of ERG) and a random dsDNA control. FUS binding RNA is then introduced to infiltrate solid condensates to produce the hollow morphology which they dub “nested asymmetric phase separation”.

Super-resolution STED microscopy is performed to elucidate and propose a model for “nested asymmetric phase separation” whereby Poly-U RNA coprecipitates with the RGG motif of FUS into a vesicle like droplet with a FUS-ERG dsDNA shell around it. A 9RA mutation of FUS-ERG abolished RNA binding and hollow condensate formation.

Experiments with the PRM-RNA system found similar results. The authors also explore a GFP-FUS-DDIT3 fusion in vivo (in U2OS cells) which forms hollow co-condensates that are also abolished with the 9RA mutation.

Revision 1. It is not clear why the authors use a different fusion construct in vivo or whether FUS-ERG would form hollow co-condensates in U2OS cells.

The authors use a phase field model to describe experimental phenomena observed in vitro and in vivo (in U2OS cells) to contribute towards our understanding of multicomponent phase separation in biological systems. They define an order parameter $\eta = \zeta_{\text{hydrophilic}} + \zeta_{\text{hydrophobic}} - \psi_{\text{critical}}$ representing the protein-DNA complex and $\phi = \zeta_{\text{hydrophilic}} - \zeta_{\text{hydrophobic}}$ representing the hollow RNA condensate and a third derived order parameter χ which represents the dsDNA concentration within the hollow condensate. In the model condensation is driven by diffusion of the dsDNA into the core of the droplet. Cahn-Hilliard dynamics (with a conserved order parameter) are used to conserve the concentration of components.

In order to test their model they fuse FUS with a Gal4 DNA binding domain and use it in combination with the “nested asymmetric phase separation” model to extract or concentrate dsDNA containing a sequence specific 1xUAS for the Gal4 DNA binding domain.

In a tertiary mixture the authors were able to separate dsDNA by DNA binding domains. They then used this method for targeted deletion of dsDNA sequences with 100% efficiency over three experiments and low (15%) off target deletion.

Revision 2. The DNA deletion experiment was not clearly described or explained in the article and supplemental material. Was a targeted nuclease used to “delete” the DNA? How was the deletion ratio determined? (See Figure 5c i and ii and Supplemental Figure 11).

The model system, use of STED and phase field model are novel, the authors were able to implement a mesoscopic empirical model based on the Cahn-Hilliard equation (with conservative order parameters) which describes complex in vitro and in vivo experiments in a coarse but predictive phase diagram.

This work represents a unique approach and multidisciplinary contribution to the study of phase separation in biomolecular complexes. The combination of the model system, super resolution microscopy and a mesoscopic phase field simulation add to our understanding of what is possible in biological and synthetic mixtures of DNA, RNA and Protein components. It builds upon and adds to previous work.

Revision 3. Live-cell evidence of hollow architectures and their functional consequences (e.g., RNA polymerase II recruitment or transcriptional activity) would significantly strengthen the biological relevance.

Reviewer #2

(Remarks to the Author)

In this work, the authors present evidence for hollow condensates formed by fusion proteins with one of the components of the fusions being derived from the FUS protein. The evidence appears to support the contention that interactions via the DNA binding domain, which is gained via fusion, drives the stabilization of hollow condensates. The authors present a mathematical model, which I have not unpacked fully because I was tripped up by confounding bits of the data which were glossed over in the model. The model appears to recapitulate the phenomenology and enable predictions. However, as alluded to above, there are issues that are glossed over here that call into the question the premise of asymmetrical phase separation. The key concern hinges on two facts: The data support a strong role for homotypic interactions being the key drivers of hollow, albeit metastable condensates that are being unmasked via an unknown mechanism, which may be DNA-specific. This is bolstered by the clear observation, albeit glossed over in the narrative, of reentrant behavior. It would appear that there is an effect reminiscent of Ostwald's rule of stages whereby the excipient, be it the specific DNA or the longer random DNA (largely ignored in the presentation) that provides either specific stabilization (microsatellite DNA) of the hollow condensate encoded by the fusion protein or via adsorption mediated effects (afforded by the longer random DNA that is not explored). The phenomenology is both clear and exciting but the mechanistic interpretation that lead to the modeling seems to gloss over several lines of evidence that point in a different direction. The sorting of a DNA and selective deletion of DNA from a library is a fantastic application that is very encouraging to read about. Overall, this work deserves to be published in Nature Communications. It is an exciting contribution with several unique elements. However, the mechanistic aspects need work as do the semantic and conceptual points. Addressing these requires more experiments and some level of rethinking of the modeling. It would also help a lot to avoid unnecessary tall claims regarding novelty. Readers should be the judges of novelty. The MS stands on its own merit without such forceful assertions. What is required right now is a precise and accurate parsing of the balance of homotypic vs. heterotypic interactions and a parsing of the contributions of site-specific vs. adsorption mediated contributions while also probing the dynamical aspects more comprehensively. The availability of a more expanded corpus of data, which will likely drive some level of rethinking of the model, will go a long way toward strengthening the manuscript and will afford greater precision with the pronouncements. Of course, my conjectures could be dead wrong, but we judge what is in front of us. Please see below for an inventory of specific concerns, which if addressed, should strengthen the manuscript and justify publication in Nature Communications.

Major comments:

1) In describing the results of Fig 1b (i) and (ii), the authors propose that FUS-ERG forms solid droplets with a random, 25 bp dsDNA. The basis for this assertion regarding the material properties was unclear from the data presented. It would appear that the term "solid" refers to a condensate that is non-hollow. If so, please clarify and / or use a different term.

2) The data presented in Fig 1c paint a different picture than what is described in the text. Going by what we see in the figure, the lowest ratio of [DNA]/[protein] at which hollow condensates appear is $(0.15 \mu\text{M} / 2 \mu\text{M})$, which translates to a ratio of 0.075 and not 0.06. Additionally, there is a reentrant behavior that is not discussed. Hollow condensates become vacuoles in larger, filled-in condensates as protein concentration increases while the DNA concentration is fixed. In these vacuolized systems, the protein and DNA always co-localize. The vacuoles shrink and the condensates are then filled in, at least at the resolution of the diffraction-limited images shown in the figure. These observations, specifically the reentrant behaviors are not included in the digitized phase diagram shown in Fig 1C (ii). The upshot is that there is a sequence involving the formation of hollow condensates, then a filling in as protein concentrations increase while keeping the DNA concentration fixed, with a shrinking of the vacuoles, and eventually a fully filled in condensate. So, hollow condensates / vacuolization shows reentrant behavior meaning that the hollow structures shrink and disappear when the protein-to-DNA ratio increases. What is perplexing is that Fig 1C (i) shows that there is a threshold protein concentration of $2 \mu\text{M}$ that needs to be crossed for the observation of hollow structures, and this is independent of DNA concentrations. This is also evident in the way Fig 1C (ii) is drawn. These observations do not square with the picture being painted in the text.

3) Overall, the content of Figure 1 points to a sort of Ostwald rule of stages being relevant in the current context. It would appear that the hollow condensates are metastable vis-a-vis the filled-in condensates. Metastable species can be kinetically / thermodynamically more accessible although this metastability seems to be tunable. Interestingly, mutations to FUS-ERG that weaken the valence of Tyr or Arg seem to weaken the metastability of hollow condensates. This seems to suggest that it is the homotypic interactions, referred to as being mediated by physical crosslinks amongst stickers (Tyr-Arg), that contribute to the metastability. In fact, the presence of the DNA-binding domain appears to unmask what is clearly an intrinsic property or preference for vesicle-like structures. Please see Fig. 3 and Fig. S5A in

<https://www.pnas.org/doi/full/10.1073/pnas.2202222119>. While Kar et al. did not, for some mysterious reason, emphasize the clear vesicular nature of the sub-saturated solution species they were studying, the relevance of this observation is germane to the current work as it raises questions about the model and the narrative being pursued.

4) The data in Fig. 2 do not provide insights on whether a higher concentration of random dsDNA or longer dsDNA - the ~300 bp system for example - will elicit similar rearrangement dynamics to what is shown in the figure. These data will be very helpful to have in order to formalize the distinction between specificity vs. a form of adsorption / wetting transition that stabilizes vacuolization and the generation of vesicular structures (not a term used by the authors) also known as hollow condensates. This is imperative because all signs point to the metastable nature of these structures. These appear to be encoded / driven by homotypic interactions and somehow unmasked by preferential, site-specific binding (microsatellite DNA) and / or adsorption / wetting (long, random DNA) and this will need to be addressed using more definitive experiments than the ones presented in Figure 2.

5) The RNA molecules are likely to be more than just passive probes. They likely alter the threshold concentration for forming hollow condensates. This should be probed by revisiting the data in Figure 1 and assessing how the RNA excipient impacts the threshold concentration for phase separation. Is the phenomenon a form of prewetting or a form of polyphasic linkage?

6) The findings reported in Ref. 14 mirror much of what is reported here, so one is hard-pressed to understand why the authors assert that their findings are a new type of molecular mechanism or why a new term needs to be coined. In fact, a recent study has also shown a cascade of wetting transitions that arise when condensates are driven by heterotypic interactions. The key difference here, which is not captured by the phenomenological model, is that homotypic interactions drive or encode the formation of the hollow condensates and that this is unmasked and stabilized by either site-specific interactions or via surface-mediated interactions (not probed experimentally because the dynamics of hollowing were not followed using the longer random DNA). Please see: <https://doi.org/10.1038/s41467-025-58736-z>. It is likely that the hollow condensates represent a form of Lifshitz point or Lifshitz line that is being unmasked by the addition of DNA.

Semantic points

1) Please update the conceptual language to reflect what we know in 2025. Condensates are not aqueous two-phase systems or vinegar droplets floating in olive oil. The phenomenology of condensation does not correspond to LLPS even for simple systems comprising a single protein in an aqueous solution. A spectrum of processes including reversible, stoichiometric binding, ion-mediated complexation, percolation, release / uptake of mobile ions and protons, solubility-driven phase separation, and changes to water activity combine in toto to the formation of condensates. The process is therefore best referred to as condensation and not LLPS.

2) Also, while it is common practice to introduce the topic with a phrase that reads XYZ is important but poorly understood, a facsimile of which we see in the second sentence of the abstract, this is jarring. It is intellectually inaccurate. Please simply delete the sentence, saving space for introducing what is being reported rather than misrepresenting the state of the field.

3) One cannot assert that a material has more gel-like properties on the basis of FRAP measurements. These measurements are useful proxies for the timescales of rearrangement. However, to be a gel is to be a network fluid, which does not necessarily have to feature slow internal dynamics. It is the networking that defines a gel. Please exercise caution with these terms.

4) Please define "phase separation capacity". Terms like heat capacity or binding capacity and even phase separation capacity have formal meanings, with these referring to response functions that quantify specific types of fluctuations. What the authors are measuring and referring to is the change in the threshold concentration for phase separation / condensation, which quantifies the driving forces for condensation, not the capacity.

Reviewer #3

(Remarks to the Author)

The hollow condensates formed by FUS-ERG is very interesting and original observation. The authors also carefully test the phase diagram, infer mechanistic insights on various constructs. The next major focus of the paper is the intracondensate organization of the phase separated components constituting the hollow condensates, and the molecular mechanism behind. This is achieved by experiments, which then enable simple mathematic modeling of the observations in a way that it recapitulates the measurements. Lastly, the authors explore the compatibility of the hollow system and the hollow FUS-Gal4 system with DNA oligos and barcodes, and demonstrate DNA substrate selectivity for the hollow condensates (and low-selectivity for homogeneous condensates), enabling DNA oligo selection. Based on my judgement, overall I see enough merit in the manuscript to be publishable in this journal after revision of the a few important points:

Major points:

- The phase separation of FUS-ERG oncofusion has been known for some time – look for example the evidence demonstrated in cellulose and in vitro by Wang et al. (2023, Nat Chem Biol). It is also known that the 25xGGAA microsatellites partition into these in vitro condensates. Such prior studies on FUS-ERG condensation are not mentioned in the present manuscript. It rather gives the impression that the authors provide the first ever evidence for this observation. The authors

must be aware of this study, as their plasmid vector is from the Pilon Li lab.

- Recombinant protein work is not reproducible, unless the protein expression and purification are described, or appropriate reference with detailed description is provided.
- It would be very nice to see the hollow nuclear condensates of the FUS-ERG oncofusion in the U2OS cells. Would it be possible?
- Condensates with polyU RNA are 10 times larger (Suppl. Fig S6) than without (15-20 vs 1.5-2.0 micrometer). It is very interesting, but I don't remember this being mentioned or discussed. Please do!
- The proposal of the conformational switch from closed to opened state of the oncofusion protein is not well-supported. More direct evidence is needed to conclude that the model holds true. Currently the results don't disprove this theory, but the observation can also be explained without this conformational change, purely based on binding events. By direct evidence, I mean proof for the closed state, i.e. binding between the RGG motif + ERG and FUS-LCD; while for the open state it should be proved that e.g. the FUS-LCD is repelled by the nucleic acid. The latter is hard to imagine, as Van Lindt et al. (2022, RNA Biology) showed that hnRNPA2-LCD – one of the well-known homologs of FUS – can bind nucleic acids by its F/YGG motifs.

Minor points:

- The Introduction chapter lacks introduction on what LLPS is. This might be better known in the readership of a specialized cellular molecular biology journal, but for a multidisciplinary journal it is advised to introduce.
- The authors often refer to "solid condensate" in the Results. "Next, biochemical assays revealed that GFP-FUS-ERG protein can form solid condensate at a concentration as low as 1 μ M in vitro (Fig. 1a). Mixing 0.6 μ M of 25- bp dsDNA containing a random sequence (referred to as 25-bp random dsDNA) and 5 μ M GFP-FUS-ERG resulted in the co-localization of both components within solid droplets (Fig. 1b(i)-(ii))." Most probably, the majority of readers will first associate to solid-state condensates without significant internal diffusion when reading this. I think the authors mean homogeneous condensates as opposed to hollow condensates, but the terminology of "solid" is confusing in this LLPS context.
- For the recombinant work, the authors centrifuged the thawed protein samples in the storage buffer to remove the aggregates. However, it is not described whether concentration was measured again on the centrifuged sample. I'd like to see protein concentration statistics on the pre- and post-centrifuged samples, if possible, as perhaps a significant drop in concentration might be experienced, therefore stock concentration may not be applicable for downstream inference of assay concentrations.
- FRAP measurements seem to be trimmed from 1800s to 500s, so FRAP recovery cannot be inferred, only the rate qualitatively. I would really like to see a plateau in the recovery for conclusive inference of the molecular behavior.
- Inference of DNA to DBD binding mainly governed by hydrophobic interaction sounds a bit far-fetched – usually electrostatic interactions play at least a comparable (if not more substantial role) in the binding of DNA. I recommend choosing a more fine-grained direct method to infer this.

Version 1:

Reviewer comments:

Reviewer #1

(Remarks to the Author)

The authors have adequately responded to reviewer comments.

Reviewer #2

(Remarks to the Author)

The authors have revised their manuscript extensively and the revisions are fully responsive to the requests that were made by me and the other reviewers. The revisions are not just extensive. They are thorough and thoughtful. The revised MS merits publication, as is, and I have no further revisions to request.

Reviewer #3

(Remarks to the Author)

Linyu Zuo, Qirui Guo and co-authors have thoroughly revised their manuscript entitled "Deciphering the molecular mechanisms of FET fusion oncoprotein-DNA hollow co-condensates" and seemingly addressed my comments to the best of their abilities.

Major points have been addressed by new experimental results, while minor points have been well-considered too, often with additional experiments. The manuscript text has been revised by the addition of new paragraphs with supplementary results and discussion as well.

These all have significantly contributed to make the story more thorough and logical, while not impacting the readability negatively. I appreciate the authors dedication, and I can now wholeheartedly recommend the manuscript for publication.

REVIEWER COMMENTS

Reviewer #1 (Remarks to the Author):

Deciphering the molecular mechanisms of FET fusion oncoprotein–DNA hollow co-condensates by Linyu Zuo et al. describes the combination of biochemical assays, super resolution imaging and mathematical modeling to understand the phase behavior of hollow co-condensates describing nested asymmetric phase separation in FUS-ERG (a fusion oncoprotein found in prostate cancer and Ewing sarcoma) and 25bp 4xGGAA dsDNA complexes which bind specifically to the DNA binding domain of ERG. The authors show that this morphology enhances data sorting enabling targeted DNA deletion in dsDNA libraries and hierarchical DNA selection.

Initially the liquid liquid phase separation morphology of FUS-ERG GFP fusions is studied by itself and in complex with fluorescently labelled 25bp 4xGGAA (specific to the DNA binding domain of ERG) and a random dsDNA control. FUS binding RNA is then introduced to infiltrate solid condensates to produce the hollow morphology which they dub “nested asymmetric phase separation”.

Super-resolution STED microscopy is performed to elucidate and propose a model for “nested asymmetric phase separation” whereby Poly-U RNA coprecipitates with the RGG motif of FUS into a vesicle like droplet with a FUS-ERG dsDNA shell around it. A 9RA mutation of FUS-ERG abolished RNA binding and hollow condensate formation.

Experiments with the PRM-RNA system found similar results. The authors also explore a GFP-FUS-DDIT3 fusion in vivo (in U20S cells) which forms hollow co-condensates that are also abolished with the 9RA mutation.

We thank the reviewer for their thoughtful and insightful summary of our manuscript.

Revision 1. It is not clear why the authors use a different fusion construct *in vivo* or whether FUS-ERG would form hollow co-condensates in U2OS cells.

We thank the reviewer for this important question. The primary aim of this manuscript is to elucidate the molecular mechanism underlying hollow condensate formation by FUS/EWS/TAF15 (FET) family fusion oncoproteins and double-stranded DNA (dsDNA), and to explore their potential biotechnological applications. Upon observing that FUS-ERG forms hollow condensates with GGAA-microsatellite-containing dsDNA *in vitro*, we investigated whether similar structures could be detected *in vivo*. Based on the Reviewer's suggestion, using confocal microscopy, we found that FUS-ERG-GFP forms small, homogeneous clusters in U2OS cells (Response Figure 1). However, due to their limited size, it remains technically challenging to determine whether these condensates exhibit a hollow architecture. Although we were unable to visualize hollow FUS-ERG condensates *in vivo*, we observed that another FET fusion oncoprotein, FUS-DDIT3-GFP, forms hollow condensates in U2OS cells—consistent with previous reports of similar spherical shells *in vivo* (Supplementary Fig. 4) ¹. Whether FUS-ERG can form hollow co-condensates in living cells remains an open and intriguing question. Future studies using super-resolution imaging technologies may help to resolve this issue.

Response Figure 1. The condensates of FET fusion oncoproteins *in vivo*.

FUS-ERG-GFP condensates in the nucleus; FUS-ERG-GFP was overexpressed in U2OS cells. Hoechst was used to stain the DNA molecules

in the nucleus (blue color).

The authors use a phase field model to describe experimental phenomena observed in vitro and in vivo (in U2OS cells) to contribute towards our understanding of multicomponent phase separation in biological systems. They define an order parameter $\eta = \zeta_{\text{hydrophilic}} + \zeta_{\text{hydrophobic}} - \psi_{\text{critical}}$ representing the protein-DNA complex and $\phi = \zeta_{\text{hydrophilic}} - \zeta_{\text{hydrophobic}}$ representing the hollow RNA condensate and a third derived order parameter χ which represents the dsDNA concentration within the hollow condensate. In the model condensation is driven by diffusion of the dsDNA into the core of the droplet. Cahn-Hilliard dynamics (with a conserved order parameter) are used to conserve the concentration of components.

In order to test their model they fuse FUS with a Gal4 DNA binding domain and use it in combination with the “nested asymmetric phase separation” model to extract or concentrate dsDNA containing a sequence specific 1xUAS for the Gal4 DNA binding domain.

In a tertiary mixture the authors were able to separate dsDNA by DNA binding domains. They then used this method for targeted deletion of dsDNA sequences with 100% efficiency over three experiments and low (15%) off target deletion.

We thank the reviewer for their thoughtful and insightful summary of our manuscript. We utilized a phase field model to accurately simulate the formation of hollow co-condensates and reveal their selective capacity for DNA.

Revision 2. The DNA deletion experiment was not clearly described or explained in the article and supplemental material. Was a targeted nuclease

used to “delete” the DNA? How was the deletion ratio determined? (See Figure 5c i and ii and Supplemental Figure 11).

We thank the reviewer for this important question and apologize for any confusion caused by the terminology. First, the term “deletion” in this context does not refer to enzymatic digestion or degradation of DNA. Rather, it denotes the selective sorting or extraction of specific DNA sequences from a heterogeneous dsDNA library by FET fusion proteins. As shown in our droplet assays (Fig. 5c(i)-(ii)), FET fusion proteins form hollow condensates preferentially with specific dsDNA sequences in the library. These sequence-specific condensates sediment upon centrifugation, resulting in a reduced concentration of the corresponding DNA in the supernatant. To quantify this effect, we performed qPCR analysis (Supplementary Fig. 13) to measure the remaining amounts of specific dsDNA sequences; Second, the “deletion ratio” is calculated as the proportion of a given dsDNA sequence that remains in the supernatant after centrifugation, relative to its original concentration in the dsDNA library before condensate formation and centrifugation. A high deletion ratio therefore indicates efficient removal/depletion of the target sequence from the library—i.e., that it was selectively incorporated into the sedimented condensates.

We have revised the relevant descriptions in both the main text and supplementary materials to clarify this terminology.

Now, they are read as (in **Results** section, **Page 18**):

“After four iterations, quantitative real-time PCR (qPCR) revealed that target-ERGDBD dsDNA and target-Gal4DBD dsDNA was removed from the library by pulling down with specific FET fusion proteins hollow condensates with an efficiency of $98.1 \pm 0.1\%$ and $100 \pm 0.0\%$, ”

in **Methods** section “DNA deletion assay”, **Page 43**:

“These sequence-specific condensates sediment upon centrifugation, resulting in a reduced concentration of the corresponding DNA in the supernatant. To quantify this effect, we performed qPCR analysis (as described in qPCR quantification) to measure the remaining amounts of specific dsDNA sequences.”

in **Methods** section “qPCR quantification”, **Page 44**:

“The “deletion ratio” is calculated as the proportion of a given dsDNA sequence that remains in the supernatant after centrifugation, relative to its original concentration in the dsDNA library before condensate formation and centrifugation. A high deletion ratio therefore indicates efficient removal/depletion of the target sequence from the library—i.e., that it was selectively incorporated into the sedimented condensates.”

The model system, use of STED and phase field model are novel, the authors were able to implement a mesoscopic empirical model based on the Cahn-Hilliard equation (with conservative order parameters) which describes complex in vitro and in vivo experiments in a coarse but predictive phase diagram.

This work represents a unique approach and multidisciplinary contribution to the study of phase separation in biomolecular complexes. The combination of the model system, super resolution microscopy and a mesoscopic phase field simulation add to our understanding of what is possible in biological and synthetic mixtures of DNA, RNA and Protein components. It builds upon and adds to previous work.

We acknowledge with appreciation the reviewer's insightful comment of our manuscript.

Revision 3. Live-cell evidence of hollow architectures and their functional consequences (e.g., RNA polymerase II recruitment or transcriptional activity) would significantly strengthen the biological relevance.

Response Figure 2. Immunofluorescence assay of RNA polymerase II (Pol II) C-terminal domain CTD in relation to FUS-DDIT3 hollow condensates *in vivo*. (a) (i) FUS-DDIT3-GFP hollow condensates did not colocalize with unphosphorylated Pol II CTD in the nucleus. (ii) Normalized intensity profiles in a(i). (b) (i) In contrast, Ser5-phosphorylated Pol II CTD was enriched at the shell of FUS-DDIT3-GFP condensates. (ii) Normalized intensity profiles in b(i). FUS-DDIT3-GFP was overexpressed in U2OS cells,

followed by fixation and incubation with antibodies (Anti-RNA polymerase II CTD repeat YSPTSPS antibody- ChIP Grade (ab26721) (Abcam), and Phospho-Rpb1 CTD (Ser5) (D9N5I) Rabbit mAb #13523 (Cell Signaling Technology)) against the RPB1 CTD overnight at 4 °C in a humidified chamber. The secondary antibody is Alexa Fluor 568 Donkey anti-Mouse IgG, Thermo Fisher, A-10037. The antibodies were a gift from the Xiong Ji lab (Peking University, China). After washing three times with PBS, cells were stained with Alexa Fluor 594-conjugated goat anti-rabbit IgG (1 : 2000) for 1.5 hours at room temperature in the dark. DNA was counterstained using Hoechst (blue) to visualize nuclei.

We agree with the reviewer's for this important comment. Our previous work demonstrated that FUS/EWS/TAF15 (FET) fusion oncoproteins are closely associated with transcriptional activation ². In response to the reviewer's suggestion, we performed additional experiments to investigate whether FUS-DDIT3 hollow condensates in the nuclei of U2OS cells can recruit RNA polymerase II (Pol II). Immunofluorescence assays revealed that unphosphorylated Pol II did not colocalize with FUS-DDIT3 hollow condensates, although it appeared to accumulate at their periphery (Revised Fig. 2a).

In contrast, Ser5-phosphorylated Pol II displayed clear colocalization with the hollow condensates (Revised Fig. 2b). Here, unphosphorylated Pol II refers to the enzyme bearing an unmodified C-terminal domain (CTD) of RPB1, while Ser5-phosphorylated Pol II refers to Pol II whose CTD heptapeptide repeats are phosphorylated at serine residue 5. These observations indicate that the phosphorylation state of Pol II CTD influences its spatial distribution relative to FUS-DDIT3 condensates, suggesting that FUS-DDIT3 hollow condensates may modulate Pol II recruitment and, by extension, transcriptional activity. Future studies will be necessary to elucidate whether a causal relationship exists between these condensates and transcriptional regulation.

Reviewer #2 (Remarks to the Author):

In this work, the authors present evidence for hollow condensates formed by fusion proteins with one of the components of the fusions being derived from the FUS protein. The evidence appears to support the contention that interactions via the DNA binding domain, which is gained via fusion, drives the stabilization of hollow condensates. The authors present a mathematical model, which I have not unpacked fully because I was tripped up by confounding bits of the data which were glossed over in the model. The model appears to recapitulate the phenomenology and enable predictions. However, as alluded to above, there are issues that are glossed over here that call into the question the premise of asymmetrical phase separation.

We thank the reviewer for their careful reading and for their valuable comments. We apologize if our manuscript did not adequately connect the assumptions of our model with the full spectrum of our experimental observations. The reviewer's feedback has prompted us to significantly revise the text to more clearly bridge the gap between our data and the model's framework. We have made modifications to the main text to address sentences that may appear unclear or overly complex. In a nutshell, we have made the following major improvements:

- (1) In the model introduction, we have replaced the detailed mathematical expressions of the order parameters with more intuitive, semantic explanations to better showcase their physical meaning and functionality in simulating hollow condensate formation. We have also revised the model assumptions and initial conditions sections to explicitly illustrate how specific experimental observations (e.g., initial dsDNA coating of condensates) directly inform our model's setup.
- (2) We have revised the parameter validation section to explicitly connect

the model's key parameters (e.g., ψ_c and b_1) to specific experimental variables (e.g., DNA binding affinity and protein hydrophobicity). This refinement aims to emphasize that our model not only reproduces our experimental observations but also reveals the essential physical principles that govern the emergence of hollow condensates.

We believe that these revisions now provide a clearer and more direct connection between the nuances of our experimental data and the underlying principles of our model. We hope these clarifications fully address the reviewer's concerns.

The key concern hinges on two facts: The data support a strong role for homotypic interactions being the key drivers of hollow, albeit metastable condensates that are being unmasked via an unknown mechanism, which may be DNA-specific. This is bolstered by the clear observation, albeit glossed over in the narrative, of reentrant behavior. It would appear that there is an effect reminiscent of Ostwald's rule of stages whereby the excipient, be it the specific DNA or the longer random DNA (largely ignored in the presentation) that provides either specific stabilization (microsatellite DNA) of the hollow condensate encoded by the fusion protein or via adsorption mediated effects (afforded by the longer random DNA that is not explored). The phenomenology is both clear and exciting but the mechanistic interpretation that lead to the modeling seems to gloss over several lines of evidence that point in a different direction. The sorting of a DNA and selective deletion of DNA from a library is a fantastic application that is very encouraging to read about. Overall, this work deserves to be published in Nature Communications. It is an exciting contribution with several unique elements. However, the mechanistic aspects need work as do the semantic and conceptual points. Addressing these requires more experiments and some level of rethinking of the modeling. It would also help a lot to avoid unnecessary tall claims regarding novelty.

Readers should be the judges of novelty. The MS stands on its own merit without such forceful assertions. What is required right now is a precise and accurate parsing of the balance of homotypic vs. heterotypic interactions and a parsing of the contributions of site-specific vs. adsorption mediated contributions while also probing the dynamical aspects more comprehensively. The availability of a more expanded corpus of data, which will likely drive some level of rethinking of the model, will go a long way toward strengthening the manuscript and will afford greater precision with the pronouncements. Of course, my conjectures could be dead wrong, but we judge what is in front of us. Please see below for an inventory of specific concerns, which if addressed, should strengthen the manuscript and justify publication in Nature Communications.

We thank the reviewer for the important and inspiring comments!

Major comments:

1) In describing the results of Fig 1b (i) and (ii), the authors propose that FUS-ERG forms solid droplets with a random, 25 bp dsDNA. The basis for this assertion regarding the material properties was unclear from the data presented. It would appear that the term "solid" refers to a condensate that is non-hollow. If so, please clarify and / or use a different term.

We thank the reviewer for this important comment. In response, we have revised the manuscript by replacing all instances of "solid condensates" with "homogeneous condensates" to ensure greater accuracy and consistency.

2) The data presented in Fig 1c paint a different picture than what is described in the text. Going by what we see in the figure, the lowest ratio of [DNA]/[protein] at which hollow condensates appear is (0.15 μ M / 2 μ M), which translates to a

ratio of 0.075 and not 0.06.

We thank the reviewer for the insightful comment. We apologize for this oversight and fully agree with the reviewer's assessment.

In response, we updated Fig. 1c(ii) (Page 27). We also revised the text in the manuscript. Now, they are read as (in **Results** section, Page 6):

“The resulting phase diagram (Fig. 1c(ii)) reveals that the DNA-to-protein molar ratio beyond a threshold number ($[DNA] / [protein] \sim 0.075$) drives the hollow co-condensate formation.”

Additionally, there is a reentrant behavior that is not discussed. Hollow condensates become vacuoles in larger, filled-in condensates as protein concentration increases while the DNA concentration is fixed. In these vacuolized systems, the protein and DNA always co-localize. The vacuoles shrink and the condensates are then filled in, at least at the resolution of the diffraction-limited images shown in the figure. These observations, specifically the reentrant behaviors are not included in the digitized phase diagram shown in Fig 1C (ii). The upshot is that there is a sequence involving the formation of hollow condensates, then a filling in as protein concentrations increase while keeping the DNA concentration fixed, with a shrinking of the vacuoles, and eventually a fully filled in condensate. So, hollow condensates / vacuolization shows reentrant behavior meaning that the hollow structures shrink and disappear when the protein-to-DNA ratio increases. What is perplexing is that Fig 1C (i) shows that there is a threshold protein concentration of 2 μM that needs to be crossed for the observation of hollow structures, and this is independent of DNA concentrations. This is also evident in the way Fig 1C (ii) is drawn. These observations do not square with the picture being painted in the text.

We thank the reviewer for this insightful comment. Reentrant phase behavior refers to the phenomenon in which variation of a single thermodynamic parameter induces two or more phase transitions, ultimately returning the system's macroscopic phase state to one that is similar or identical to its original state ³. In biomolecular condensates, two distinct forms of reentrant behavior have been documented. The first follows a “two-phase → one-phase → two-phase” sequence, producing an hourglass-shaped phase diagram ⁴. For example, proteins such as FUS, Sox2, and Brd4 form condensates at both low and high salt concentrations, yet remain well mixed at intermediate KCl concentrations ⁵. The second form follows a “one-phase → two-phase → one-phase” sequence, resulting in a closed-loop phase diagram ⁴. This behavior has been observed in certain RNA–protein co-condensates composed of arginine-rich peptides and homopolymeric single-stranded RNA, where condensates form only at intermediate RNA concentrations, but not at low or high RNA concentrations ^{6,7}.

In Fig. 1c, we examined the conditions required for hollow co-condensate formation using *in vitro* droplet assays. We combined 25-bp 4× GGAA dsDNA at concentrations of 0, 0.15, 0.3, 0.6, 1.2, and 2.4 μM with GFP-FUS-ERG protein at concentrations of 0.25, 0.5, 2, 5, and 10 μM (Fig. 1c(i)). When the DNA concentration was fixed at 0.15 μM, hollow condensates formed only at a protein concentration of approximately 2 μM, with no hollow condensates observed at either lower or higher protein concentrations. At protein concentrations below 2 μM, no condensates were detected, whereas at concentrations above 2 μM, the hollow structures became filled, yielding homogeneous condensates. This non-monotonic dependence on protein concentration about the hollow structure formation is a hallmark of reentrant phase behavior ^{6,7}.

The resulting phase diagram (Fig. 1c(ii)) shows that hollow co-condensate formation occurs when the DNA-to-protein molar ratio exceeds a threshold of

approximately 0.075. This threshold arises because the protein alone can form homogeneous condensates at concentrations above $\sim 2 \mu\text{M}$. Notably, increasing the DNA concentration from $0.15 \mu\text{M}$ to $2.4 \mu\text{M}$ did not alter the protein concentration threshold for hollow structure formation, which remained fixed at $\sim 2 \mu\text{M}$, indicating that it is independent of DNA concentration. The molecular basis of this reentrant phase behavior in FUS-ERG – dsDNA co-condensation remains to be elucidated and will be the subject of future investigation.

In response, we updated Fig. 1c(ii) (Page 27). We also updated the manuscript. Now, they are read as (in Results section, Page 6):

“ We investigated the conditions required for hollow co-condensate formation through in vitro droplet assays, combining 25-bp $4\times$ GGAA dsDNA at concentrations of 0, 0.15, 0.3, 0.6, 1.2, and $2.4 \mu\text{M}$ with GFP-FUS-ERG protein at concentrations of 0.25, 0.5, 2, 5, and $10 \mu\text{M}$ (Fig. 1c(i)). When the DNA concentration was fixed at $0.15 \mu\text{M}$, hollow condensates formed only at a protein concentration of approximately $2 \mu\text{M}$, with no hollow condensates observed at either lower or higher protein concentrations. At protein concentrations below $2 \mu\text{M}$, no condensates were detected, whereas at concentrations above $2 \mu\text{M}$, the hollow structures became filled, yielding homogeneous condensates. This non-monotonic dependence on protein concentration about the hollow structure formation is a hallmark of reentrant phase behavior^{6,7}.

The resulting phase diagram (Fig. 1c(ii)) reveals that the DNA-to-protein molar ratio beyond a threshold number ($[\text{DNA}] / [\text{protein}] \sim 0.075$) drives the hollow co-condensate formation. This threshold arises because the protein alone can form homogeneous condensates at concentrations above $\sim 2 \mu\text{M}$. Notably, increasing the DNA concentration from $0.15 \mu\text{M}$ to $2.4 \mu\text{M}$ did not alter the threshold of protein concentration for hollow structure formation, which remained fixed at $\sim 2 \mu\text{M}$. The molecular basis of this reentrant phase behavior

in these hollow condensate remains to be elucidated and will be the subject of future investigation.”

3) Overall, the content of Figure 1 points to a sort of Ostwald rule of stages being relevant in the current context. It would appear that the hollow condensates are metastable vis-a-vis the filled-in condensates. Metastable species can be kinetically / thermodynamically more accessible although this metastability seems to be tunable. Interestingly, mutations to FUS-ERG that weaken the valence of Tyr or Arg seem to weaken the metastability of hollow condensates. This seems to suggest that it is the homotypic interactions, referred to as being mediated by physical crosslinks amongst stickers (Tyr-Arg), that contribute to the metastability. In fact, the presence of the DNA-binding domain appears to unmask what is clearly an intrinsic property or preference for vesicle-like structures. Please see Fig. 3 and Fig. S5A in <https://www.pnas.org/doi/full/10.1073/pnas.2202222119>. While Kar et al. did not, for some mysterious reason, emphasize the clear vesicular nature of the sub-saturated solution species they were studying, the relevance of this observation is germane to the current work as it raises questions about the model and the narrative being pursued.

We thank the reviewer for this important comment. We totally agreed with the Reviewer’s comment, and revised the manuscript. Now, they are read as (in **Discussion** section, **Page 20**):

“The hollow condensates observed in Fig. 1 are consistent with the Ostwald rule of stages, suggesting that these structures may represent metastable intermediates along the pathway toward more thermodynamically stable states. This metastability appears tunable, in part through homotypic interactions mediated by physical crosslinks between sticker residues, such as tyrosine–arginine contacts (Fig. 1d–e). The finding that dsDNA containing GGAA

microsatellites promotes hollow condensate formation suggests that the DNA-binding domain can reveal and stabilize an intrinsic propensity for such architectures⁸, either through site-specific recognition (Fig. 2a) or surface-mediated interactions (Supplementary Fig. 2c). These observations are consistent with previous reports of vesicle-like structures formed by FUS proteins under subsaturated conditions⁹, supporting the existence of a latent vesicular phase. We further speculate that these hollow condensates may correspond to a Lifshitz point or fall along a Lifshitz line, with DNA acting as a trigger to unmask this phase behavior. This possibility opens new avenues for probing the thermodynamic and kinetic landscape governing condensate morphology.”

4) The data in Fig. 2 do not provide insights on whether a higher concentration of random dsDNA or longer dsDNA - the ~300 bp system for example - will elicit similar rearrangement dynamics to what is shown in the figure. These data will be very helpful to have in order to formalize the distinction between specificity vs. a form of adsorption / wetting transition that stabilizes vacuolization and the generation of vesicular structures (not a term used by the authors) also known as hollow condensates. This is imperative because all signs point to the metastable nature of these structures. These appear to be encoded / driven by homotypic interactions and somehow unmasked by preferential, site-specific binding (microsatellite DNA) and / or adsorption / wetting (long, random DNA) and this will need to be addressed using more definitive experiments than the ones presented in Figure 2.

We thank the reviewer for this insightful comment. In response, we conducted additional experiments to investigate the impact of DNA length on hollow condensate formation (Response Figure 3). Specifically, we repeated the experiment shown in Supplementary Fig. 2a using random DNA substrates of varying lengths—199, 120, and 60 base pairs (bp)—in place of the original 306-

bp sequence (Response Figure 3a). Notably, these DNA substrates lacked GGAA motifs. We observed that only the 306-bp random DNA supported hollow condensate formation with GFP-FUS-ERG, whereas shorter DNA fragments failed to do so, suggesting a critical length threshold of approximately 120 bp is required for this process.

Response Figure 3. GFP-FUS-ERG condensates with random dsDNA in different lengths. (a) GFP-FUS-ERG condensates with random dsDNA in different lengths. 5 μ M GFP-FUS-ERG mixed with 60-bp random dsDNA (i), 120-bp GGAA dsDNA (ii), 120-bp GGAA dsDNA (iii). These DNA

substates' concentrations are all same as 10 ng/ μ L, which is consistent to the mass concentration of 0.6 μ M 25-bp DNA used in the main manuscript. All *in vitro* droplet assays were executed under physiological conditions, specifically 40 mM Tris-HCl (pH = 7.5), 150 mM KCl, 2 mM MgCl₂, 1 mM DTT and 0.2 mg/mL BSA, with thorough mixing and a 30-minute incubation period prior to imaging, unless otherwise indicated. (b) 306-bp random dsDNA substrates can transfer into the solid FUS-ERG condensates, fail to induce the hollow co-condensate formation. Time course of hollow co-condensate formation of 5 μ M GFP-FUS-ERG mixed with 10ng/ μ L AlexaFluor647-labeled 306-bp random dsDNA at 0-min (i), 30-min (ii), 60-min (iii), and 90-min (iv). At the 0-minute time point, we injected the 306-bp random dsDNA.

We next repeated the experiment shown in Fig. 2a, substituting the 25-bp 4 \times GGAA dsDNA with 306-bp random DNA (Response Figure 3b). To monitor the temporal dynamics of condensate remodeling, we performed a two-step assay. Homogeneous condensates were first formed using GFP-FUS-ERG (as in Fig. 1a), followed by the addition of 306-bp random DNA (Response Figure 3b(i), time 0). The system was then imaged continuously for 90 minutes (Response Figure 3b). Strikingly, even after 90 minutes (Response Figure 3b(iv)), the DNA infiltrated the condensates and colocalized with protein, but no hollow condensate formation was observed—contrasting sharply with the outcome in Fig. 2a. These findings highlight a key mechanistic distinction between site-specific binding by microsatellite DNA and non-specific adsorption or wetting by long random DNA ⁸, which may differentially stabilize hollow condensates. Exploring this distinction represents an interesting avenue for future research.

In response, we added Response Figure 3b into Supplementary Fig. 2, and updated the figure caption (Page 70).

We also revised the manuscript. Now, they are read as (in **Results** section, **Page 8**):

“Second, we repeated the experiment shown in Fig. 2a, substituting the 25-bp 4× GGAA dsDNA with 306-bp random dsDNA (Supplementary Fig. 2c). Interestingly, even after 90 minutes (Supplementary Fig. 2c(iv)), the dsDNA infiltrated the condensates and colocalized with protein, but no hollow condensate formation was observed—contrasting sharply with the outcome in Fig. 2a. These findings demonstrate the spontaneous transfer of dsDNA containing GGAA microsatellites into the condensates accompanying hollow co-condensate formation. These findings also highlight a key mechanistic distinction between site-specific binding by microsatellite DNA and non-specific adsorption or wetting by long random DNA⁸, which may differentially stabilize hollow condensates. Exploring this distinction represents an interesting avenue for future research.”

5) The RNA molecules are likely to be more than just passive probes. They likely alter the threshold concentration for forming hollow condensates. This should be probed by revisiting the data in Figure 1 and assessing how the RNA excipient impacts the threshold concentration for phase separation. Is the phenomenon a form of prewetting or a form of polyphasic linkage?

We thank the reviewer for this insightful comment. In the main text, our focus was primarily on characterizing the hydrophobic and hydrophilic properties of the hollow condensate shell by examining the spatial distribution of RNA (Fig. 3a and c). Thus, we did not initially address the role of RNA in the formation of hollow condensates. In response to the reviewer’s suggestion, we conducted an *in vitro* droplet assay to assess the impact of RNA on hollow condensate formation. Specifically, we mixed 5 μM SNAP-FUS-ERG with increasing concentrations (0.15 to 3.6 μM) of 25-bp 4× GGAA dsDNA. Control samples contained no RNA (Response Figure 4a), while experimental samples were

supplemented with 0.3 μM of 50-nt polyU RNA (Response Figure 4b). All samples were imaged after a 30-minute incubation at room temperature.

Response Figure 4. Effect of RNA on hollow condensate formation. (a)-(b) *In vitro* droplet assay of hollow condensate with or without RNA. In each condition, SNAP-FUS-ERG was 5 μM and Cy3-labeled 25-bp 4x GGAA DNA was 0.15 to 3.6 μM . (a) Control groups, no RNA was added. (b) Experimental groups, 0.3 μM Cy5-labeled 50-nt PolyU RNA was added. All samples were

imaged after 30-minute incubation at room temperature. The working buffer contains 40 mM Tris-HCl (pH 7.5), 150 mM KCl, 2 mM MgCl₂, 1 mM DTT and 0.2 mg/mL BSA. (c) Area analysis of all samples shown in (a) and (b). The area was derived from the signal of Cy3-labeled 25-bp 4× GGAA DNA. N indicates the number of independent condensates analyzed under each condition. In box plots, the black line denotes the median, box edges represent the 25th and 75th percentiles, whiskers indicate the range excluding outliers, and outliers are shown as individual dots (•). Statistical comparisons were performed using unpaired two-tailed t-tests. P value style: GP: 0.1234 (ns), 0.0332 (*), 0.0021 (**), 0.0002 (***), < 0.0001 (****).

The results demonstrated that hollow condensates formed under all DNA concentrations tested, regardless of the presence or absence of RNA. Notably, We compared the size of condensates of the two groups (with and without RNA added) under various DNA concentration conditions, and found that within the range we tested, regardless of the DNA concentration, the area of hollow condensates (including the hollow region) mixed with RNA added was significantly smaller. Collectively, these findings indicate that RNA primarily influences the volume fraction of hollow condensates after their formation, rather than altering the threshold conditions required for their initiation.

To determine whether RNA enrichment within hollow condensates reflects a prewetting phenomenon or polyphasic linkage, we performed a new *in vitro* droplet assay. The following is the idea for the experimental design: both prewetting and wetting describe the binding behavior of molecules on the surface of another object. When the molecular concentration is lower than its saturation concentration for phase separation, the molecules will bind to the surface and undergo prewetting. In this condition, the molecules will spread on the surface to form a thin layer, and some areas will further aggregate to form a thick layer. When the molecular concentration exceeds its saturation

concentration, the molecules will undergo phase separation and form partial wetting or complete wetting on the surface. It should be noted that both prewetting and wetting involve two independent coexisting phases between the molecules and the original surface¹⁰. Therefore, in our experiments, we maintained constant initial concentrations of 5 μM SNAP-FUS-ERG and 0.6 μM Cy3-labeled 25-bp 4 \times GGAA DNA, while varying the initial concentration of Cy5-labeled 50-nt polyU RNA, and monitored the spatial distribution of RNA signals as a function of its concentration to determine whether it follows the above-mentioned prewetting or wetting behavior (Response Figure 5a).

Response Figure 5. Effect of RNA concentration on the spatial

distribution of RNA on the hollow condensate. (a) *In vitro* droplet assay of hollow condensate with different concentration of RNA. In each condition, SNAP-FUS-ERG was 5 μM and Cy3-labeled 25-bp 4 \times GGAA DNA was 0.6 μM . Cy5-labeled 50-nt PolyU RNA was added from 0 to 0.6 μM . The contrast of RNA signal was adjust in the condition with 0, 0.01, 0.03 and 0.06 μM RNA to display the details of the images more clearly. (b) Intensity plot of Cy3-labeled 25-bp 4 \times GGAA DNA and Cy5-labeled 50-nt PolyU RNA in (a). (i) belongs to the condition with 0.06 μM Cy5-labeled 50-nt PolyU RNA, while (ii) belongs to the condition with 0.6 μM Cy5-labeled 50-nt PolyU RNA. (c) *In vitro* droplet assay of Cy5-labeled 50-nt PolyU RNA alone with varying concentration. All samples were imaged after 30-minute incubation at room temperature. The working buffer contains 40 mM Tris-HCl (pH 7.5), 150 mM KCl, 2 mM MgCl_2 , 1 mM DTT and 0.2 mg/mL BSA.

The experiment revealed a notable phenomenon: the spatial distribution of RNA changed with its initial concentration. When the initial RNA concentration was ≤ 0.3 μM (e.g. 0.06 μM shown in Response Figure 5b(i)), RNA predominantly accumulated within the lumen of hollow condensates. By contrast, at 0.6 μM RNA, the fluorescence peak of RNA colocalized with the Cy3-labeled 25-bp 4 \times GGAA DNA signal at the condensate shell (Response Figure 5b(ii)). This indicates that at this condition, RNA and the shell of the original hollow condensate are no longer two independent components, but instead show a state of partial miscibility. Furthermore, It should be noted that RNA cannot form condensates on its own, whether within the concentration range covered by the above experiments (0-0.6 μM) or when the concentration was further increased to 3.6 μM (Response Figure 5c), and we did not observe regions with low RNA concentration within RNA-enriched areas at the hollow region inside hollow condensates. These phenomena strongly suggest that the enrichment of RNA does not result from prewetting or wetting, but is most likely

recruitment of RNA into the hollow region through an unknown mechanism.

Interestingly, at low RNA concentrations, RNA accumulates inside the hollow region, with its intensity peak positioned far from the shell (Response Figure 5b(i)). This indicates that RNA is repelled by the shell. When RNA concentration is increased, the RNA intensity peak colocalizes with the shell (Response Figure 5b(ii)), suggesting that RNA is now attracted by the shell. This indicates that the interaction between RNA and the inner face is not fixed; instead, it changes with increasing RNA concentration, which also differs from the consistent interaction strength between molecules and interfaces observed in processes such as prewetting and wetting. These phenomena warrant further investigation in future studies.

6) The findings reported in Ref. 14 mirror much of what is reported here, so one is hard-pressed to understand why the authors assert that their findings are a new type of molecular mechanism or why a new term needs to be coined.

We thank the reviewer for this insightful comment. We totally agreed that the findings reported in Ref. 14 mirror much of what is reported here. That is why we carefully read this paper, and cited this reference (Ref. 14) in Discussion section. This reference found that in the *in vitro* reconstitution of mitochondrial transcription machinery, the formation of hollow co-condensates can play a crucial role in regulating transcription rates¹¹. They also conducted coarse-grained Monte Carlo simulations to reproduce the distribution of major components in hollow mt-transcriptional condensates, which suggested that the vesicles formed *in vitro* are the result of dynamical arrest. Although their work reveals one mechanism about hollow condensate formation, several critical questions remain to be further explored. These include the detailed molecular interactions, conformational changes and movement patterns of biomolecules (protein, RNA and DNA) during the transition of condensates from a

homogeneous state to a hollow state, as well as the thermodynamic driving forces underlying this phenomenon. The central finding of this study is that RNA transcribed within transcriptional condensates plays a pivotal role in driving hollow condensate formation. While DNA may serve as a structural scaffold, the combination of DNA and transcription factors alone is insufficient to produce hollow condensates in mitochondria, and this phenomenon has not been observed *in vivo*.

In our case, dsDNA containing GGAA microsatellites can directly induce hollow condensate formation with the FET family fusion oncoprotein FUS-ERG. The hollow condensate is DNA and protein two-component condensates. Using biochemical assays, super-resolution imaging, and mathematical modeling, we try to understand the molecule mechanism and revealed the formation of hollow condensates through the process of nested asymmetric phase separation. Thus, building upon the work presented in Ref. 14, our study supplements the understanding of the transition of condensates from a homogeneous state to a hollow state—including discussions on the details of biomolecular interactions, conformational changes, movements and thermodynamic driving forces—thereby adding new insights into the formation mechanism of hollow condensates. We updated the manuscript to clearly mention this point.

In fact, a recent study has also shown a cascade of wetting transitions that arise when condensates are driven by heterotypic interactions. The key difference here, which is not captured by the phenomenological model, is that homotypic interactions drive or encode the formation of the hollow condensates and that this is unmasked and stabilized by either site-specific interactions or via surface-mediated interactions (not probed experimentally because the dynamics of hollowing were not followed using the longer random DNA). Please see: <https://doi.org/10.1038/s41467-025-58736-z>. It is likely that the hollow condensates represent a form of Lifshitz point or Lifshitz line that is being

unmasked by the addition of DNA.

We thank the reviewer for this insightful comment. We totally agreed with the Reviewer's comment, and updated the manuscript. Now, they are read as (in **Discussion** section, **Page 20**):

“The hollow condensates observed in Fig. 1 are consistent with the Ostwald rule of stages, suggesting that these structures may represent metastable intermediates along the pathway toward more thermodynamically stable states. This metastability appears tunable, in part through homotypic interactions mediated by physical crosslinks between sticker residues, such as tyrosine–arginine contacts (Fig. 1d–e). The finding that dsDNA containing GGAA microsatellites promotes hollow condensate formation suggests that the DNA-binding domain can reveal and stabilize an intrinsic propensity for such architectures ⁸, either through site-specific recognition (Fig. 2a) or surface-mediated interactions (Supplementary Fig. 2c). These observations are consistent with previous reports of vesicle-like structures formed by FUS proteins under subsaturated conditions ⁹, supporting the existence of a latent vesicular phase. We further speculate that these hollow condensates may correspond to a Lifshitz point or fall along a Lifshitz line, with DNA acting as a trigger to unmask this phase behavior. This possibility opens new avenues for probing the thermodynamic and kinetic landscape governing condensate morphology.”

Semantic points

1) Please update the conceptual language to reflect what we know in 2025. Condensates are not aqueous two-phase systems or vinegar droplets floating in olive oil. The phenomenology of condensation does not correspond to LLPS even for simple systems comprising a single protein in an aqueous solution. A spectrum of processes including reversible, stoichiometric binding, ion-

mediated complexation, percolation, release / uptake of mobile ions and protons, solubility-driven phase separation, and changes to water activity combine in toto to the formation of condensates. The process is therefore best referred to as condensation and not LLPS.

We thank the reviewer for this insightful comment. We fully agree with the suggestion and have replaced all instances of “LLPS” with “biomolecular condensates” throughout the manuscript to improve clarity and consistency for a multidisciplinary readership.

2) Also, while it is common practice to introduce the topic with a phrase that reads XYZ is important but poorly understood, a facsimile of which we see in the second sentence of the abstract, this is jarring. It is intellectually inaccurate. Please simply delete the sentence, saving space for introducing what is being reported rather than misrepresenting the state of the field.

We thank the reviewer for this valuable comment. As suggested, we have removed the second sentence from the abstract.

3) One cannot assert that a material has more gel-like properties on the basis of FRAP measurements. These measurements are useful proxies for the timescales of rearrangement. However, to be a gel is to be a network fluid, which does not necessarily have to feature slow internal dynamics. It is the networking that defines a gel. Please exercise caution with these terms.

We thank the reviewer for this important comment. In response, we have replaced the term “gel-like” with “slow internal dynamics” to more accurately reflect the observed material properties.

Now, they are read as (in **Results** section, **Page 7**):

“The molecular dynamics of GFP-FUS-ERG in hollow co-condensates were slower compared to homogeneous condensates, suggesting more slower internal dynamics inside the shell region of hollow co-condensates.”

4) Please define "phase separation capacity". Terms like heat capacity or binding capacity and even phase separation capacity have formal meanings, with these referring to response functions that quantify specific types of fluctuations. What the authors are measuring and referring to is the change in the threshold concentration for phase separation / condensation, which quantifies the driving forces for condensation, not the capacity.

We thank the reviewer for this important comment. In response, we have replaced the term “phase separation capacity” with “threshold concentration for condensation,” which more precisely quantifies the driving forces underlying condensate formation.

Reviewer #3 (Remarks to the Author):

The hollow condensates formed by FUS–ERG is very interesting and original observation. The authors also carefully test the phase diagram, infer mechanistic insights on various constructs. The next major focus of the paper is the intracondensate organization of the phase separated components constituting the hollow condensates, and the molecular mechanism behind. This is achieved by experiments, which then enable simple mathematic modeling of the observations in a way that it recapitulates the measurements. Lastly, the authors explore the compatibility of the hollow system and the hollow FUS-Gal4 system with DNA oligos and barcodes, and demonstrate DNA substrate selectivity for the hollow condensates (and low-selectivity for homogeneous condensates), enabling DNA oligo selection. Based on my judgement, overall I see enough merit in the manuscript to be publishable in this journal after revision of the a few important points:

We thank the reviewer for their thoughtful summary and positive assessment.

Major points:

- The phase separation of FUS–ERG oncofusion has been known for some time – look for example the evidence demonstrated in cellulo and in vitro by Wang et al. (2023, Nat Chem Biol). It is also known that the 25xGGAA microsatellites partition into these in vitro condensates. Such prior studies on FUS–ERG condensation are not mentioned in the present manuscript. It rather gives the impression that the authors provide the first ever evidence for this observation. The authors must be aware of this study, as their plasmid vector is from the Pulong Li lab.

We thank the reviewer for this important comment and apologize for the lack of clarity in the original text. We are deeply grateful to the Pulong Li Laboratory

(Tsinghua University, China) for providing the plasmid vectors and the U2OS human cancer cell line used in our experiments (**Methods**). Li and colleagues have previously demonstrated that the FUS–ERG fusion protein undergoes phase separation both in cells and *in vitro*¹². We do not claim to be the first to report this observation, and we have clarified this point in the revised **Results** section.

Now, they are read as (in **Results** section, **Page 5**):

“Next, biochemical assays revealed that GFP-FUS-ERG protein can form homogeneous condensate at a concentration as low as 1 μM in vitro (Fig. 1a), consistent with previous findings¹².”

- Recombinant protein work is not reproducible, unless the protein expression and purification are described, or appropriate reference with detailed description is provided.

We thank the reviewer for this important comment. In response, we have added the appropriate reference² in the **Methods** section, which includes a detailed description.

Now, they are read as (in **Methods** section, **Page 38**):

“Proteins were expressed and purified using affinity chromatography as described in our earlier work².”

- It would be very nice to see the hollow nuclear condensates of the FUS–ERG oncofusion in the U2OS cells. Would it be possible?

We thank the reviewer for this important comment. Using confocal microscopy, we found that FUS-ERG-GFP forms small, homogeneous clusters in U2OS cells (Response Figure 6). However, due to their limited size, it remains

technically challenging to determine whether these condensates exhibit a hollow architecture. Although we were unable to visualize hollow FUS-ERG condensates *in vivo*, we observed that another FET fusion oncoprotein, FUS-DDIT3-GFP, forms hollow condensates in U2OS cells—consistent with previous reports of similar spherical shells *in vivo* (Supplementary Fig. 4) ¹. Whether FUS-ERG can form hollow co-condensates in living cells remains an open and intriguing question. Future studies using super-resolution imaging technologies may help to resolve this issue.

Response Figure 6. The condensates of FET fusion oncoproteins *in vivo*.

FUS-ERG-GFP condensates in the nucleus; FUS-ERG-GFP was overexpressed in U2OS cells. Hoechst was used to stain the DNA molecules in the nucleus (blue color).

- Condensates with polyU RNA are 10 times larger (Suppl. Fig S6) than without (15-20 vs 1.5-2.0 micrometer). It is very interesting, but I don't remember this being mentioned or discussed. Please do!

We thank the reviewer for this important comment. In our experiments, we found that FUS-RGG3 (FUS⁴⁷²⁻⁵⁰⁴), as shown in Supplementary Fig. 8, does not undergo liquid-liquid phase separation (LLPS) on its own. We also revisited the study by Banerjee and colleagues ¹³, which reported that protamine (PRM)—an arginine-rich, disordered nucleoprotein similar to the RGG motifs of FUS—can form hollow condensates with RNA. Notably, their thermodynamic

phase diagram (Fig. 2A) indicates that PRM alone is insufficient to drive LLPS, similar to our observation with FUS-RGG3.

While we cannot comment on the precise mechanism underlying PRM–RNA hollow condensate formation, we investigated this point in our system. Specifically, we observed that 5 μM GFP–FUS–ERG and 25-bp 4 \times GGAA dsDNA form hollow condensates with an average diameter of $2.1 \pm 0.6 \mu\text{m}$ (N = 137) (Revised Fig. 7b–c). In contrast, 5 μM GFP–FUS–ERG alone forms smaller, homogeneous condensates with an average diameter of $1.4 \pm 0.2 \mu\text{m}$ (N = 69) (Revised Fig. 7a and c). The mechanistic basis for why the presence of DNA leads to the formation of larger hollow condensates remains an open question and warrants further investigation.

Response Figure 7. The presence of DNA leads to the formation of larger hollow condensates. (a) GFP-FUS-ERG can undergo LLPS in the concentration of 5 μM (Data from Fig. 1a); (b) 5 μM GFP-FUS-ERG mixed with 0.6 μM 25-bp 4 \times GGAA dsDNA. dsDNA was labeled with AlexaFluor647. (Data from Fig. 1b(iv)). All *in vitro* droplet assays were executed under physiological conditions, specifically 40 mM Tris-HCl (pH = 7.5), 150 mM KCl, 2 mM MgCl₂,

1 mM DTT and 0.2 mg/mL BSA, with thorough mixing and a 30-minute incubation period prior to imaging, unless otherwise indicated. (c) Boxplot of condensate diameter for the condition in a and b. The total number N examined over one-time *in vitro* droplet experiments. For the boxplot, the red bar represents median. The bottom edge of the box represents 25th percentiles, and the top is 75th percentiles. Most extreme data points are covered by the whiskers except outliers. The '+' symbol is used to represent the outliers. Statistical significance was analyzed using unpaired t test for two groups. P value: two-tailed; p value style: GP: 0.1234 (ns), 0.0332 (*), 0.0021 (**), 0.0002 (***), <0.0001 (****). Confidence level: 95%.

- The proposal of the conformational switch from closed to opened state of the oncofusion protein is not well-supported. More direct evidence is needed to conclude that the model holds true. Currently the results don't disprove this theory, but the observation can also be explained without this conformational change, purely based on binding events. By direct evidence, I mean proof for the closed state, i.e. binding between the RGG motif + ERG and FUS-LCD; while for the open state it should be proved that e.g. the FUS-LCD is repelled by the nucleic acid. The latter is hard to imagine, as Van Lindt et al. (2022, RNA Biology) showed that hnRNPA2-LCD – one of the well-known homologs of FUS – can bind nucleic acids by its F/YGG motifs.

We thank the reviewer for this important comment. To test the conformational states of GFP-FUS-ERG—namely the “closed” and “open” configurations regulated by dsDNA containing GGAA microsatellites (Fig. 3e(i))—we adopted the experimental strategy proposed by the reviewer. Specifically, we divided the full-length GFP-FUS-ERG into two separate modules: GFP-FUS-LCD and RGG-ERG (comprising the RGG motif and ERG domain). SDS-PAGE and EMSA analyses confirmed that both proteins were purified to high homogeneity and that RGG-ERG retained sequence-specific DNA-binding activity

(Response Figure 8b). Notably, GFP-FUS-LCD did not exhibit any detectable DNA-binding, even at very high concentrations (Response Figure 8a). This observation contrasts with the findings reported in Van Lindt et al. (2022, RNA Biology) ¹⁴, which demonstrated that the LCD of hnRNPA2 can interact with a broad range of nucleic acids, including both RNA and DNA.

Response Figure 8. Biochemical assays for GFP-FUS-LCD and RGG-ERG. *In vitro* purified GFP-FUS-LCD (a) and RGG-ERG (b). (i) Schematic representation of the protein constructs; (ii) SDS-PAGE analysis confirming protein purity; (iii) Electrophoretic mobility shift assay (EMSA) performed on a 1.2% agarose gel. DNA substrates were labeled with Quasar670 and imaged using an Amersham Typhoon RGB system (635 nm excitation, Cy5 670BP30 emission filter).

To test whether FUS-ERG can adopt a closed conformation mediated by interactions between the FUS-LCD and the ERG domain, we performed an *in vitro* droplet assay. GFP-FUS-LCD and RGG-ERG proteins were incubated either individually or in combination at equimolar concentrations for 30 minutes at room temperature prior to imaging (Response Figure 9a(i)). GFP-FUS-LCD alone did not form condensates, while RGG-ERG readily formed droplets on its own. Upon mixing, the two proteins exhibited clear co-localization of fluorescent signals (Response Figure 9b(i)), indicating direct interactions between GFP-FUS-LCD and RGG-ERG. These results strongly support the presence of a closed state driven by intramolecular or intermolecular contacts between the FUS-LCD and ERG domain (Fig. 3e(i)).

Response Figure 9. High concentration of dsDNA containing GGAA motifs can inhibit the interaction between GFP-FUS-LCD and RGG-ERG.

(a) *In vitro* droplet assays of GFP-FUS-LCD, RGG-ERG, and mixtures of the two proteins with or without dsDNA. (i) 20 μ M GFP-FUS-LCD only, 20 μ M

RGG-ERG only and 20 μM GFP-FUS-LCD mixed with 20 μM RGG-ERG. (ii)-(iii) The addition of varying concentrations of 25-bp 4 \times GGAA DNA (ii) or 25-bp Random DNA (iii) to a system containing 20 μM GFP-FUS-LCD mixed with 20 μM RGG-ERG. (b) Normalized intensity profiles from conditions in (a). (i) 20 μM GFP-FUS-LCD mixed with 20 μM RGG-ERG. (ii)-(iii) Conditions containing 20 μM GFP-FUS-LCD, 20 μM RGG-ERG and varying concentrations of 25-bp 4 \times GGAA DNA (ii) or 25-bp Random DNA (iii). In this set of experiments, RGG-ERG was prepared by mixing unlabeled protein with AlexaFluor647-labeled protein at a ratio of 100:1. In all experimental conditions, the proteins and DNA were first mixed, and then imaged after incubation at room temperature for 30 minutes. The working buffer containing 40 mM Tris-HCl (pH 7.5), 150 mM KCl, 2 mM MgCl_2 , 1 mM DTT and 0.2 mg/mL BSA.

To determine whether double-stranded DNA (dsDNA) containing GGAA motifs can induce a conformational transition of FUS-ERG from a closed to an open state, we performed a droplet assay in which GFP-FUS-LCD and RGG-ERG were mixed at equimolar concentrations and incubated with increasing amounts of 25-bp 4 \times GGAA DNA or control 25-bp random DNA (Response Figure 9a(ii–iii)). At a low concentration of 4 \times GGAA DNA (0.3 μM), GFP-FUS-LCD continued to co-localize with RGG-ERG. However, at or above 0.6 μM , this co-localization was abolished (Response Figure 9b(ii)). In contrast, random DNA failed to disrupt co-localization even at concentrations as high as 2.4 μM (Response Figure 9b(iii)). These results indicate that only high concentrations of GGAA-containing dsDNA can displace GFP-FUS-LCD from RGG-ERG condensates, likely by disrupting the interaction between the FUS-LCD and ERG domain. This observation aligns with the previous work¹⁵, which showed that high-affinity DNA disrupts the interaction between EWS-LCD and FLI1-DBD within condensates. Collectively, these data strongly support that GGAA

motif-containing dsDNA induces a transition of FUS-ERG from a closed to an open state. Moreover, our results demonstrate that control dsDNA lacking GGAA motifs does not elicit this conformational change. Together with the droplet assay results from Response Figure 9 and Fig. 1b, our findings support that the closed-to-open transition of FUS-ERG is a critical prerequisite for the formation of hollow condensates.

In response, we used Response Figure 8-9 as new Supplementary Fig. 6-7, and updated these figures' caption (Page 74-76).

We also revised the manuscript. Now, they are read as (in **Results** section, Page 10-12):

“To test the hypothesis of conformational switch from closed to opened state of GFP-FUS-ERG in Fig. 3e, we divided the full-length GFP-FUS-ERG into two separate modules: GFP-FUS-LCD and RGG-ERG (comprising the RGG motif and ERG domain). SDS-PAGE and EMSA analyses confirmed that both proteins were purified to high homogeneity and that RGG-ERG retained sequence-specific DNA-binding activity (Supplementary Fig. 6a). Notably, GFP-FUS-LCD did not exhibit any detectable DNA-binding, even at very high concentrations (Supplementary Fig. 6b).

To test whether FUS-ERG can adopt a closed conformation mediated by interactions between the FUS-LCD and the ERG domain, we performed an in vitro droplet assay. GFP-FUS-LCD and RGG-ERG proteins were incubated either individually or in combination at equimolar concentrations for 30 minutes at room temperature prior to imaging (Supplementary Fig. 7a(i)). GFP-FUS-LCD alone did not form condensates, while RGG-ERG readily formed droplets on its own. Upon mixing, the two proteins exhibited clear co-localization of fluorescent signals (Supplementary Fig. 7b(i)), indicating direct interactions between GFP-FUS-LCD and RGG-ERG. These results strongly support the presence of a closed state driven by intramolecular or intermolecular contacts between the FUS-LCD and ERG domain (Fig. 3e(i)).

To determine whether dsDNA containing GGAA motifs can induce a conformational transition of FUS-ERG from a closed to an open state, we performed an in vitro droplet assay in which GFP-FUS-LCD and RGG-ERG were mixed at equimolar concentrations and incubated with increasing amounts of 25-bp 4× GGAA DNA or control 25-bp random DNA (Supplementary Fig. 7a(ii–iii)). At a low concentration of 4× GGAA DNA (0.3 μM), GFP-FUS-LCD continued to co-localize with RGG-ERG. However, at or above 0.6 μM, this co-localization was abolished (Supplementary Fig. 7b(ii)). In contrast, random DNA failed to disrupt co-localization even at concentrations as high as 2.4 μM (Supplementary Fig. 7b(iii)). These results indicate that only high concentrations of GGAA-containing dsDNA can displace GFP-FUS-LCD from RGG-ERG condensates, likely by disrupting the interaction between the FUS-LCD and ERG domain. This observation aligns with the previous work¹⁵, which showed that high-affinity DNA disrupts the interaction between EWS-LCD and FLI1-DBD within condensates. Collectively, these data strongly support that GGAA motif-containing dsDNA induces a transition of FUS-ERG from a closed to an open state. Moreover, our results demonstrate that control dsDNA lacking GGAA motifs does not elicit this conformational change. Together with the droplet assay results from Supplementary Fig. 7 and Fig. 1b, our findings support that the closed-to-open transition of FUS-ERG is a critical prerequisite for the formation of hollow condensates.”

Minor points:

- The Introduction chapter lacks introduction on what LLPS is. This might be better known in the readership of a specialized cellular molecular biology journal, but for a multidisciplinary journal it is advised to introduce.

We thank the reviewer for this important comment. We have clarified the concept of LLPS in the **Introduction** section.

Now, they are read as (in **Introduction** section, **Page 3**):

“Phase separation is a fundamental physicochemical process by which biomolecules—such as proteins and nucleic acids—spontaneously demix from the surrounding solution (the dilute phase) to form concentrated, mesoscale condensates (the dense phase) within cells¹⁶⁻²⁰. These biomolecular condensates are dynamic, membraneless compartments composed of multiple components and are formed independently of lipid bilayers. Their assembly is primarily driven by two mechanisms: interactions mediated by intrinsically disordered regions (IDRs) of proteins²¹, and multivalent interactions among modular macromolecules²².”

- The authors often refer to “solid condensate” in the Results. “Next, biochemical assays revealed that GFP-FUS-ERG protein can form solid condensate at a concentration as low as 1 μ M in vitro (Fig. 1a). Mixing 0.6 μ M of 25- bp dsDNA containing a random sequence (referred to as 25-bp random dsDNA) and 5 μ M GFP-FUS-ERG resulted in the co-localization of both components within solid droplets (Fig. 1b(i)-(ii)).” Most probably, the majority of readers will first associate to solid-state condensates without significant internal diffusion when reading this. I think the authors mean homogeneous condensates as opposed to hollow condensates, but the terminology of “solid” is confusing in this LLPS context.

We thank the reviewer for this important comment. In response, we have revised the manuscript by replacing all instances of “solid condensates” with “homogeneous condensates” to ensure greater accuracy and consistency.

- For the recombinant work, the authors centrifuged the thawed protein samples in the storage buffer to remove the aggregates. However, it is not described

whether concentration was measured again on the centrifuged sample. I'd like to see protein concentration statistics on the pre- and post-centrifuged samples, if possible, as perhaps a significant drop in concentration might be experienced, therefore stock concentration may not be applicable for downstream inference of assay concentrations.

Response Figure 10. Comparison of protein concentration between the uncentrifuged protein stock and the supernatants after centrifugation. (a) Raw data of A_{280}/cm measured by Nanodrop. **(b)** Protein concentrations calculated based on the results in a and the protein extinction coefficient at 280 nm ($108530 \text{ M}^{-1}\text{cm}^{-1}$). Independent experiments were repeated 3 times respectively ($n = 3$). Statistical significance was analyzed using an unpaired t-test for two groups. P value: two-tailed. Confidence level: 95%.

We thank the reviewer for this important comment. To address this point, we compared protein stocks subjected to centrifugation (12,000 rpm for 10 minutes at 4°C) with those that were not centrifuged. Protein concentrations were measured using a Nanodrop spectrophotometer. The results revealed no

significant difference between the uncentrifuged samples and the supernatants post-centrifugation, indicating that the centrifugation step does not affect the protein concentration and thus does not influence the experimental outcomes reported in our manuscript (Response Figure 10).

- FRAP measurements seem to be trimmed from 1800s to 500s, so FRAP recovery cannot be inferred, only the rate qualitatively. I would really like to see a plateau in the recovery for conclusive inference of the molecular behavior.

We thank the reviewer for this important comment. Upon re-examination of our original FRAP data, we confirmed that the measurement duration was 500 seconds, not 1800 seconds. In the **Methods** section, we stated: “*The samples were prepared as described and the mixed samples were incubated at room temperature for 30 minutes.*” We believe the reviewer may have interpreted this 30-minute incubation (equivalent to 1,800 seconds) as the measurement duration. Nonetheless, we appreciate the reviewer’s feedback, as it prompted us to recognize that the normalized fluorescence intensity of GFP–FUS–ERG did not reach saturation in the original measurements. We have therefore repeated the FRAP experiments with an extended measurement time of 1,800 seconds. Specifically, 5 μM GFP–FUS–ERG, with or without 0.6 μM 25-bp 4 \times GGAA dsDNA, was mixed and loaded into 384-well plates. Following a 30-minute incubation at room temperature, FRAP measurements were performed either at the center of homogeneous condensates (without dsDNA) or at the shell of hollow condensates (with 0.6 μM 25-bp 4 \times GGAA dsDNA). The updated results, presented in Response Figure 11, show that within the initial 0–500 second interval, fluorescence recovery at the shell of hollow condensates was lower than at the center of homogeneous condensates, consistent with the data in the original Supplementary Fig. 3b(ii). We have replaced Supplementary Fig. 3b(ii) with these new FRAP results obtained over the extended 1,800-second measurement period.

Response Figure 11. FRAP analysis of GFP–FUS–ERG in homogeneous versus hollow condensates. Homogeneous condensates were formed by 5 μM GFP–FUS–ERG alone, whereas hollow condensates were assembled by mixing 5 μM GFP–FUS–ERG with 0.6 μM 25-bp 4 \times GGAA dsDNA in a 384-well plate. After incubation at room temperature for 30 minutes, fluorescence in the center of homogeneous condensates or at the shell of hollow condensates was photobleached, followed by recording of fluorescence recovery over an additional 30 minutes. FRAP recovery curves are shown in green for GFP–FUS–ERG in the center of homogeneous condensates and in orange for GFP–FUS–ERG at the shell of hollow condensates. The total number of condensates analyzed (N) is indicated. Error bars represent mean \pm s.d.

- Inference of DNA to DBD binding mainly governed by hydrophobic interaction sounds a bit far-fetched – usually electrostatic interactions play at least a comparable (if not more substantial role) in the binding of DNA. I recommend choosing a more fine-grained direct method to infer this.

We thank the reviewer for their valuable comments. We fully agree that electrostatic forces are crucial for the initial binding of double-stranded DNA (dsDNA) to proteins. However, Our model's assumption regarding the preferential localization of dsDNA (χ) within hydrophobic regions aims to capture a different aspect: not the energetics of the initial dsDNA-DBD binding event itself, but rather the subsequent spatial organization of the entire dsDNA-FUS-ERG complex within the condensates. As is shown in the text, the ERG DBD, to which dsDNA binds (Fig. 3e(i)), exhibits significant hydrophobicity compared to other FUS-ERG regions, as shown by Nile-red staining (Supplementary Fig. 5b). Therefore, once dsDNA is bound to this hydrophobic DBD, the resulting dsDNA-FUS-ERG complex effectively presents an amphiphilic character. This emergent amphiphilicity is crucial for our phase-field model, which is based on the Ohta-Kawasaki framework designed for such systems. It allows us to model how these complexes then self-organize into microphase-separated structures, such as the observed hollow condensates. The "hydrophobic preference" in our model thus describes the effective partitioning of the dsDNA-bound ERG domains into hydrophobic microenvironments within the condensate. While initial dsDNA-DBD binding is multifaceted, the subsequent stable arrangement of these complexes to form the hollow internal surface strongly correlates with the hydrophobic nature of the DBD-rich regions. The success of our model—in accurately reproducing hollow condensate morphology (Fig. 4), the system's response to parameter variations (Supplementary Fig. 9), and its predictive power (Supplementary Fig. 10) validates that this effective treatment captures the key physics driving the mesoscale architecture.

As for the considerations to model the DNA-protein interactions with fine-grained modes, we also fully agree that a fine-grained molecular dynamics simulation would indeed be the appropriate tool to dissect the detailed binding energies at the dsDNA-DBD interface. However, our current phase-field model

is designed to bridge molecular properties to macroscopic condensate organization, a scale at which explicitly modeling atomistic binding details becomes computationally prohibitive and is beyond the model's intended scope.

To further probe the hydrophobic and hydrophilic properties of the ERG DBD–DNA complex, we conducted electrophoretic mobility shift assays (EMSAs) to assess the DNA-binding behavior of full-length SNAP-FUS-ERG and the truncated RGG-ERG protein (validated in Response Figure 12) across varying salt concentrations. Binding of SNAP-FUS-ERG to double-stranded DNA (dsDNA) containing GGAA motifs remained largely unaffected as KCl concentrations increased from 150 mM to 500 mM (Response Figure 12a). In contrast, RGG-ERG—lacking the FUS-LCD—exhibited efficient DNA binding at 150 mM KCl, but binding was markedly reduced at 500 mM KCl (Response Figure 12b). These results suggest that ERG DBD–DNA binding is primarily electrostatically driven, while the FUS-LCD contributes a local hydrophobic environment that stabilizes the complex under high-salt conditions. This potential hydrophobic contribution presents an interesting direction for future investigation.

We hope these clarifications adequately address the reviewer's concerns.

Response Figure 12. EMSA assays. EMSA of SNAP-FUS-ERG (a) and RGG-ERG (b) under varying salt ion concentrations. (i) 150 mM KCl. (ii) 500 mM KCl. DNA substrates were labeled with Quasar670 and imaged using an Amersham Typhoon RGB system (635 nm laser with Cy5 670BP30 filter).

REFERENCES

- 1 Davis, R. B., Kaur, T., Moosa, M. M. & Banerjee, P. R. FUS oncofusion protein condensates recruit mSWI/SNF chromatin remodeler via heterotypic interactions between prion-like domains. *Protein Sci* **30**, 1454-1466, doi:10.1002/pro.4127 (2021).
- 2 Zuo, L. *et al.* Loci-specific phase separation of FET fusion oncoproteins

- promotes gene transcription. *Nature Communications* **12**, 1491, doi:10.1038/s41467-021-21690-7 (2021).
- 3 Portz, B. & Shorter, J. Biochemical Timekeeping Via Reentrant Phase Transitions. *J Mol Biol* **433**, 166794, doi:<https://doi.org/10.1016/j.jmb.2020.166794> (2021).
- 4 Pappu, R. V., Cohen, S. R., Dar, F., Farag, M. & Kar, M. Phase Transitions of Associative Biomacromolecules. *Chemical Reviews* **123**, 8945-8987, doi:10.1021/acs.chemrev.2c00814 (2023).
- 5 Krainer, G. *et al.* Reentrant liquid condensate phase of proteins is stabilized by hydrophobic and non-ionic interactions. *Nature Communications* **12**, 1085, doi:10.1038/s41467-021-21181-9 (2021).
- 6 Alshareedah, I. *et al.* Interplay between Short-Range Attraction and Long-Range Repulsion Controls Reentrant Liquid Condensation of Ribonucleoprotein-RNA Complexes. *J Am Chem Soc* **141**, 14593-14602, doi:10.1021/jacs.9b03689 (2019).
- 7 Banerjee, P. R., Milin, A. N., Moosa, M. M., Onuchic, P. L. & Deniz, A. A. Reentrant Phase Transition Drives Dynamic Substructure Formation in Ribonucleoprotein Droplets. *Angew Chem Int Edit* **56**, 11354-11359, doi:10.1002/anie.201703191 (2017).
- 8 Erkamp, N. A. *et al.* Differential interactions determine anisotropies at interfaces of RNA-based biomolecular condensates. *Nature Communications* **16**, 3463, doi:10.1038/s41467-025-58736-z (2025).
- 9 Kar, M. *et al.* Phase-separating RNA-binding proteins form heterogeneous distributions of clusters in subsaturated solutions. *Proceedings of the National Academy of Sciences* **119**, e2202222119, doi:doi:10.1073/pnas.2202222119 (2022).
- 10 Zhao, X., Bartolucci, G., Honigmann, A., Jülicher, F. & Weber, C. A. Thermodynamics of wetting, prewetting and surface phase transitions with surface binding. *New Journal of Physics* **23**, 123003, doi:10.1088/1367-2630/ac320b (2021).
- 11 Feric, M. *et al.* Mesoscale structure–function relationships in mitochondrial transcriptional condensates. *Proceedings of the National Academy of Sciences* **119**, e2207303119, doi:doi:10.1073/pnas.2207303119 (2022).
- 12 Wang, Y. *et al.* Dissolution of oncofusion transcription factor condensates for cancer therapy. *Nat Chem Biol* **19**, 1223-1234, doi:10.1038/s41589-023-01376-5 (2023).
- 13 Alshareedah, I., Moosa, M. M., Raju, M., Potoyan, D. A. & Banerjee, P. R. Phase transition of RNA-protein complexes into ordered hollow condensates. *P Natl Acad Sci USA* **117**, 15650-15658, doi:10.1073/pnas.1922365117 (2020).
- 14 Van Lindt, J. *et al.* F/YGG-motif is an intrinsically disordered nucleic-acid binding motif. *RNA Biol* **19**, 622-635, doi:10.1080/15476286.2022.2066336 (2022).

- 15 Selig, E. E. *et al.* Phase separation of the oncogenic fusion protein EWS::FLI1 is modulated by its DNA-binding domain. *Proceedings of the National Academy of Sciences* **122**, e2221823122, doi:doi:10.1073/pnas.2221823122 (2025).
- 16 Banani, S. F., Lee, H. O., Hyman, A. A. & Rosen, M. K. Biomolecular condensates: organizers of cellular biochemistry. *Nat Rev Mol Cell Bio* **18**, 285-298, doi:10.1038/nrm.2017.7 (2017).
- 17 Berry, J., Brangwynne, C. P. & Haataja, M. Physical principles of intracellular organization via active and passive phase transitions. *Reports on Progress in Physics* **80**, doi:ARTN 046601 10.1088/1361-6633/aaa61e (2018).
- 18 Alberti, S. & Dormann, D. Liquid-Liquid Phase Separation in Disease. *Annual Review of Genetics, Vol 53* **53**, 171-194, doi:10.1146/annurev-genet-112618-043527 (2019).
- 19 Boeynaems, S. *et al.* Protein Phase Separation: A New Phase in Cell Biology. *Trends in Cell Biology* **28**, 420-435, doi:10.1016/j.tcb.2018.02.004 (2018).
- 20 Shin, Y. & Brangwynne, C. P. Liquid phase condensation in cell physiology and disease. *Science* **357**, doi:ARTN eaaf4382 10.1126/science.aaf4382 (2017).
- 21 Uversky, V. N. Intrinsically disordered proteins in overcrowded milieu: Membrane-less organelles, phase separation, and intrinsic disorder. *Current Opinion in Structural Biology* **44**, 18-30, doi:10.1016/j.sbi.2016.10.015 (2017).
- 22 Li, P. L. *et al.* Phase transitions in the assembly of multivalent signalling proteins. *Nature* **483**, 336-U129, doi:10.1038/nature10879 (2012).

REVIEWER COMMENTS

Reviewer #1 (Remarks to the Author):

The authors have adequately responded to reviewer comments.

We thank the reviewer for the insightful comment of our manuscript.

Reviewer #2 (Remarks to the Author):

The authors have revised their manuscript extensively and the revisions are fully responsive to the requests that were made by me and the other reviewers. The revisions are not just extensive. They are thorough and thoughtful. The revised MS merits publication, as is, and I have no further revisions to request.

We thank the reviewer for the insightful comment of our manuscript.

Reviewer #3 (Remarks to the Author):

Linyu Zuo, Qirui Guo and co-authors have thoroughly revised their manuscript entitled “Deciphering the molecular mechanisms of FET fusion oncoprotein–DNA hollow co-condensates” and seemingly addressed my comments to the best of their abilities.

Major points have been addressed by new experimental results, while minor points have been well-considered too, often with additional experiments. The manuscript text has been revised by the addition of new paragraphs with supplementary results and discussion as well.

These all have significantly contributed to make the story more thorough and logical, while not impacting the readability negatively. I appreciate the authors dedication, and I can now wholeheartedly recommend the manuscript for publication.

We thank the reviewer for the insightful comment of our manuscript.